# Deciphering bat influenza H18N11 infection dynamics in male Jamaican fruit bats on a single-cell level

Susanne Kessler [1,2,18], Bradly Burke [3,18], Geoffroy Andrieux [2,4], Jan Schinköthe [5], Lea Hamberger[1,2], Johannes Kacza[6], Shijun Zhan[3], Clara Reasoner[3], Taru S. Dutt [7], Maria Kaukab Osman[1,2,8,9], Marcela Henao-Tamayo [7], Julian Staniek [9,10,11], Jose Francisco Villena Ossa[11,12], Dalit T. Frank[3], Wenjun Ma[13], Reiner Ulrich[5], Toni Cathomen [2,11,12], Melanie Boerries [2,4,14], Marta Rizzi [10,11,15,16], Martin Beer [17], Martin Schwemmle [1,2], Peter Reuther [1,2], Tony Schountz [3] ✉ & Kevin Ciminski [1,2] ✉

Jamaican fruit bats (*Artibeus jamaicensis*) naturally harbor a wide range of viruses of human relevance. These infections are typically mild in bats, suggesting unique features of their immune system. To better understand the immune response to viral infections in bats, we infected male Jamaican fruit bats with the bat-derived influenza A virus (IAV) H18N11. Using comparative single-cell RNA sequencing, we generated single-cell atlases of the Jamaican fruit bat intestine and mesentery. Gene expression profiling showed that H18N11 infection resulted in a moderate induction of interferon-stimulated genes and transcriptional activation of immune cells. H18N11 infection was predominant in various leukocytes, including macrophages, B cells, and NK/T cells. Confirming these findings, human leukocytes, particularly macrophages, were also susceptible to H18N11, highlighting the zoonotic potential of this bat-derived IAV. Our study provides insight into a natural virus-host relationship and thus serves as a fundamental resource for future in-depth characterization of bat immunology.

Bats belong to the order Chiroptera, the second largest order of mammals, comprising ~1400 species and accounting for about 20% of mammalian species worldwide[1]. In recent years, bats have received increasing attention as reservoirs for numerous emerging zoonotic viruses, including certain ebolaviruses, Marburg virus, Nipah virus, Hendra virus, and MERS and SARS coronaviruses[2–11]. These viruses often cause severe disease and death in humans, whereas infections in bats are often innocuous, even when associated with high viral replication[12]. Several recent in vitro studies suggested that the combination of an enhanced innate immune response and the ability to limit extensive inflammation may lead to a state of viral tolerance in

bats[12–16]. However, these findings have yet to be corroborated in an in vivo system[17]. The Jamaican fruit bat (*Artibeus jamaicensis*) is a frugivorous bat species whose main habitat ranges from the rainforests of southern Mexico to the Caribbean and northern regions of South America[18]. Their relatively simple housing and husbandry practices and an annotated genome make Jamaican fruit bats an attractive model system for in vivo bat studies[17,19].

Since 2010, several genome sequences of a previously undescribed influenza A virus (IAV), classified as H18N11, have been exclusively identified in rectal swab samples from Neotropical fruit bats (*Artibeus*)[20,21], suggesting that they constitute the natural reservoirs for

this bat IAV. Unlike known IAV, H18N11 infects cells by using major histocompatibility complex class II (MHC-II) molecules[22,23]. In all species characterized to date, MHC-II is constitutively expressed on professional antigen-presenting cells (APCs), including monocytes, macrophages, dendritic cells, and B cells[24]. We have previously shown that H18N11 replication is predominantly detected in the gastrointestinal tract and APC-rich gut-associated lymphoid tissues of experimentally infected Jamaican fruit bats[25]. While these findings suggest that H18N11 targets immune cells−in contrast to conventional IAVs which mainly target gut or respiratory epithelia−the specific cell type(s) infected by H18N11 in bats remains unknown[26−29].

To date, in-depth in vivo characterization of bats at a cellular and molecular level has been largely hampered by the lack of available reagents and suitable tools. Single-cell RNA sequencing (scRNA-seq) has emerged as a powerful next-generation sequencing approach that allows antibody-independent characterization and quantification of cell types and the study of biological processes in heterogeneous cell populations[30]. Recent studies have shown that scRNA-seq can also be used to study bat immunology in-depth, including during viral infection[31,32]. In this study, we performed scRNA-seq of H18N11-infected Jamaican fruit bats to study this natural virus-host interaction on a single-cell level. We decipher the cellular landscape of the bat intestine and mesentery, and provide initial insights into the transcriptional changes of individual cell types in response to infection. Using a combination of scRNA-seq with in situ hybridization, we reveal that H18N11 preferentially infects different types of leukocytes, including macrophages, B cells, NK/T cells, and less frequently intestinal epithelial cells. Consistent with these findings, we show that H18N11 can also infect human leukocytes and that human macrophages support efficient viral replication. Overall, this study characterizes the tropism and infection dynamics of Jamaican fruit bats with the bat IAV H18N11 on a single-cell level, thereby providing fundamental resources for the study of bat immunology.

## Results

### Infection of bats with H18N11

Our single-cell RNA sequencing (scRNA-seq) analysis was based on a comparison of uninfected and infected bats. We infected a group of six bats with wild-type (WT) H18N11, another group of six animals with a mutant H18N11 variant lacking the non-structural protein 1 (ΔNS1), and four bats served as uninfected controls. The viral NS1 is a potent antagonist of the host innate immune signaling pathways, promoting efficient viral replication[33,34]. Disruption of NS1 by targeted mutagenesis of the C-terminal effector domain is thus a common approach to study the immunomodulatory function of NS1 (Supplementary Fig. 1a)[35]. Virus shedding from oronasally inoculated bats was confirmed by titration of rectal swab material collected at 2 and 7 days post-infection (dpi; Supplementary Fig. 1b). We detected infectious virus in all WT H18N11-infected bats up to 7 dpi, but not in rectal swabs from ΔNS1-infected bats, suggesting that viral growth of the NS1-deficient H18N11 variant was attenuated. However, both WT and ΔNS1-infected bats seroconverted with H18N11-specific antibodies at 9 dpi (Supplementary Fig. 1c).

### Cellular landscape of the bat intestine and mesentery

At 3 and 9 dpi, representing an early and a late time point of the acute infection[25], we euthanized three animals from each group and generated single-cell transcriptomes for each bat intestine and mesentery (Fig. 1a). We first compiled an inventory of the cell types present in the intestine and mesentery of Jamaican fruit bats, as previous findings by ourselves and others identified these as the main target organs of H18N11 infection[21,25]. To maximize the identification of rare cell populations and to generate a complete immunological landscape of the intestine and mesentery, we analyzed scRNA-seq data for these organs from both infected and uninfected bats. After quality control and

removal of low-quality cells, 33,974 intestinal cells were profiled (Supplementary Fig. 2a−c). In total, 24 distinct cell clusters were identified using Louvain clustering on the top 20 principal components and visualized using uniform manifold approximation and projection (UMAP) (Fig. 1b and Supplementary Fig. 3a). These intestinal cell clusters were manually annotated based on the RNA expression profile of highly conserved mammalian marker genes previously reported in the literature for human and mouse (Supplementary Fig. 4 and Supplementary Data 1)[36,37]. This identified intestinal epithelial cell types, including enterocytes, crypt and goblet cells, and endothelial cells, including vascular endothelial cells (VECs) and lymphatic endothelial cells (LECs), as well as enteric glia cells. The remaining clusters represented immune cells of the myeloid and lymphoid lineages. The five myeloid cell types present in the intestine included neutrophils, mast cells, type 1 conventional dendritic cells (cDC1s), macrophages, and *Cd14*+ macrophages. Neutrophils were defined as positive for *Cd14* and *Cd177*, a specific neutrophil activation marker, and additionally expressed *Csf3r*, which encodes the granulocyte colony-stimulating factor receptor, and the major neutrophil chemokine receptor *Cxcr2*[38,39]. Mast cells were identified by the expression of the gene for the high-affinity immunoglobulin (Ig)E receptor alpha subunit *Fcer1a*, and the genes encoding the mast/stem cell growth factor receptor *Kit* and the mast cell carboxypeptidase A *Cpa3*[40]. The cDC1 cell signature was based on enrichment for the lineage-specific transcription factor genes *Batf3* and *Irf8* and the receptor genes *Clec9a* and *Xcr1*[41−43]. Both macrophage subsets had the canonical *Cd68* and *Cd163* markers[44], but transcripts for *Cd14* and *Slc11a1* and high levels of the pro-inflammatory cytokine gene *Il1b* were specific for *Cd14*+ macrophages. Intestinal lymphoid populations included B cells and their terminal differentiation stage of plasma cells, intestinal intraepithelial lymphocytes (IELs), and type 3 innate lymphoid cells (ILC3s). Three B-cell clusters expressed the canonical marker gene *Cd19*, the co-receptor *Cd79b* along with the gene encoding the lineage-specific transcription factor *Pax5*[45]. *Aicda*, encoding activation-induced cytidine deaminase (AID), was exclusively expressed in one B-cell population (cluster 13), identifying B cells undergoing class switching and affinity maturation[46]. Plasma cells were identified by expression of the *Irf4* transcription factor, *Tnfrsf17* encoding the B-cell maturation antigen, and the Ig linker *Jchain*[45,47]. IELs had enriched copies of the T cell receptor (TCR) alpha, beta, and gamma chain genes, as well as *Cd3d*, *Cd8a*, and *Prf1*, but lacked expression of *Cd8b*[48,49]. ILC3s lacked TCR genes and were characterized by the expression of *Rorc*, a lineage-specific master transcription factor[50], and the genes encoding the cytokines IL17A, IL17F, and IL22[51].

Although all of the above immune cell types were present in each of the five bat groups, their relative proportions varied, reflecting changes in the intestinal immune cell fractions after infection (Fig. 1c, Supplementary Fig. 3b, and Supplementary Data 2). Among myeloid cells, cDC1s and both macrophage subsets were enriched in ΔNS1-infected bats at 3 dpi compared to uninfected controls. Neutrophils, which were the most abundant immune cells in the intestine of uninfected bats, were enriched in two of three WT H18N11-infected animals at 3 dpi and in all bats of the ΔNS1 group at 9 dpi. Among lymphoid cells, we observed that H18N11 infection resulted in a substantial increase in B cells, which were enriched in two of three samples from WT-infected animals at 3 and 9 dpi. In contrast, IELs, which accounted for up to 25% of all immune cells in the intestine of uninfected control bats, decreased up to fourfold after infection (Fig. 1c, Supplementary Fig. 3b, and Supplementary Data 2).

A total of 20,600 mesenteric cells passed quality control and were further profiled (Supplementary Fig. 2d−f). In total, 18 cell clusters were identified using Louvain clustering on the top 20 principal components and visualized using UMAP (Fig. 1d and Supplementary Fig. 5a). These mesenteric cell clusters were manually annotated using canonical cell expression markers (Supplementary Fig. 6 and

Supplementary Data 1). Non-immune cell types in the mesentery included progenitor cells (PCs), enterocytes, and undifferentiated hematopoietic stem cells and multipotent progenitors (HSC/MPPs), as well as a population of cells (cluster 2) that exhibited an expression profile that could not be attributed to a specific cell type. Because this unknown cell cluster also had relatively low read and feature counts, suggesting a potential overpopulation of empty droplets, we excluded these cells from further analysis.

We defined six myeloid cell types in the mesentery, including neutrophils, monocytes, macrophages, *Cd14*⁺ macrophages, cDC1s, and mature lymph node interdigitating DCs (IDCs). The mesenteric populations of neutrophils, macrophages, *Cd14*⁺ macrophages, and cDC1s expressed the abovementioned canonical marker genes. Monocyte populations were identified by high transcript levels of the

monocyte chemokine receptor *Ccr2* together with *Cd33*, *Cd52*, *Lilra5*, and *Self*[52–56]. IDCs were characterized by the maturation markers *Ccr7*, *Il4i1*, and *Lamp3* and the gene encoding the lymph node migration factor *Fscn1*[57–60]. Mesenteric lymphoid cell types included B cells, naïve T cells, and NK/T cells. Two B-cell clusters expressed the specific markers *Cd19*, *Cd79b*, and *Pax5* together with *Jchain*, indicating a mixed population of naïve and mature B cells and plasma cells. Three cell populations (clusters 6, 8, and 13) exhibited a T cell-like expression profile with enrichment for the TCR and its signaling subunits *Lck* and *Zap70*. Importantly, despite low *Cd4* expression, these clusters did not segregate into *Cd4*⁺ or *Cd8a*⁺ populations, but co-expressed both co-receptor genes of the TCR[31]. However, because these three clusters were also enriched in *Eomes*, which encodes a key transcriptional activator of NK cell maturation[61] as well as the killer lectin-like receptor

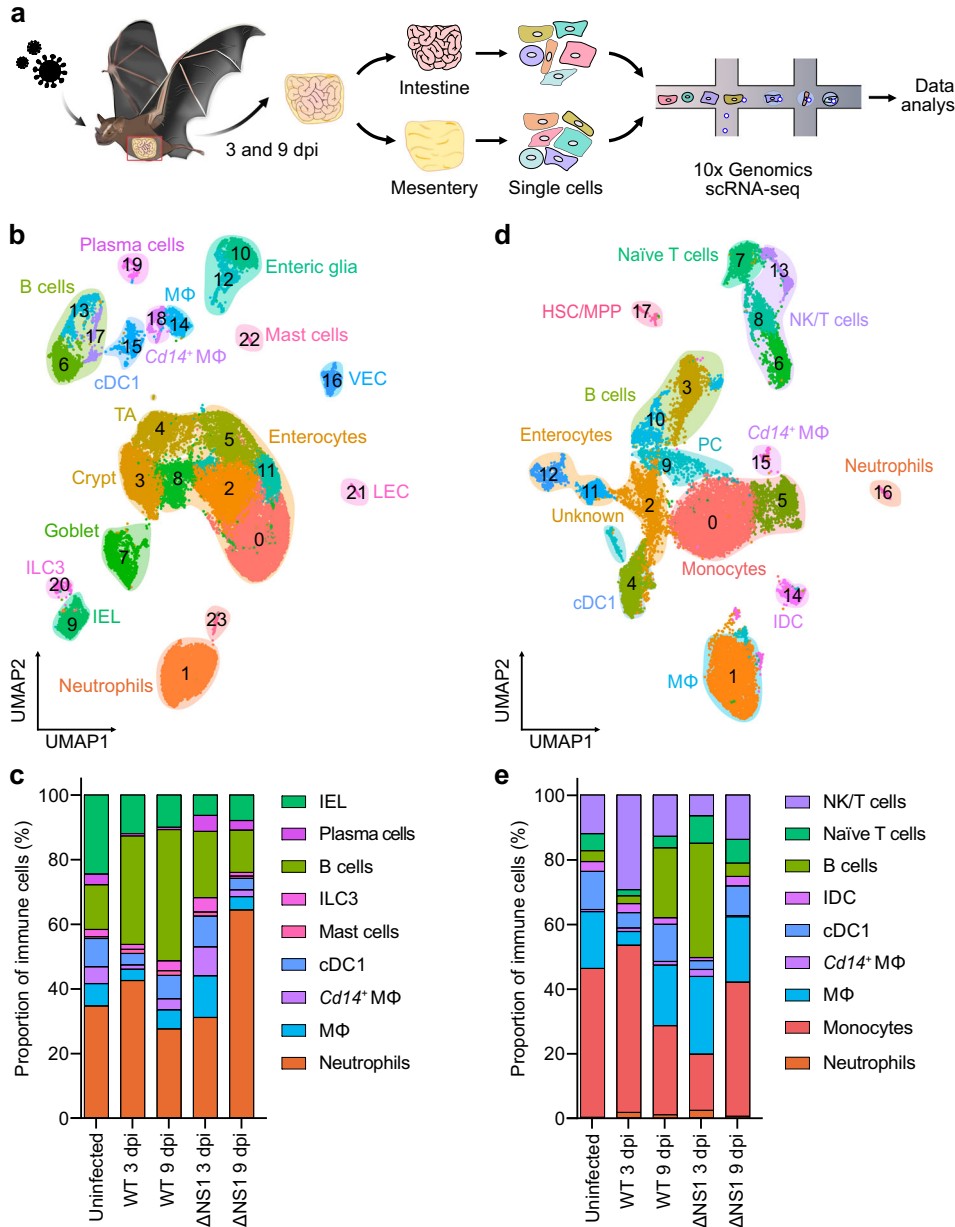

**Fig. 1 | Single-cell landscapes of the Jamaican fruit bat intestine and mesentery. a** Overview of the study design. **b** UMAP embedding of intestinal cells. Each cell cluster is indicated by a Louvain clustering ID and annotated cell types are superimposed. MΦ, macrophages; *Cd14*⁺ MΦ; *Cd14*⁺ macrophages. **c** Bar graph showing the proportion of the indicated immune cell types relative to all immune cells in the

intestine of uninfected, WT H18N11, or ΔNS1-infected bats at the indicated time points. **d** UMAP embedding of mesenteric cells as described in (**b**). **e** Bar graph showing the proportion of the indicated immune cell types relative to all immune cells in the mesentery of each bat group at the indicated time points. Source data are provided as a Source Data file.

(KLR) genes *Klrd1*, *Klrg1*, and *Klrk1* and the natural cytotoxicity receptor (NCR) genes *Ncr1* and *Ncr3*, they were classified as NK/T cells. Consistent with previous reports on other bat species, no transcripts for killer-cell immunoglobulin-like receptor (KIR) genes were found[62,63], and we noted the lack of expression of CD16 and CD56, which are canonical NK cell markers in humans[64,65]. The expression of the TCR genes, together with high levels of *Ccr7* and *Il7r* identified a population of naïve T cells.

Having defined cell types, we then determined the relative frequencies of mesenteric immune cells between the different bat groups. Although all mesenteric immune cell types were present in each group, we observed both sample-wise differences between individual bats within the same group as well as infection-related changes in cell-type frequencies between groups (Fig. 1e, Supplementary Fig. 5b, and Supplementary Data 2). Whereas no significant changes were observed among myeloid cells between the different bat groups, the total number of B cells was substantially increased in two of three bats in the WT-infected group at 9 dpi. Moreover, NK/T cells, which were the most abundant lymphoid cells in the mesentery of uninfected bats, were enriched in WT-infected animals at 3 dpi (Fig. 1e, Supplementary Fig. 5b, and Supplementary Data 2).

### Infection induces genes involved in the innate immune response

To analyze the global transcriptomic changes associated with infection, we performed hallmark gene-set analysis on intestinal and mesenteric cells from infected bats compared to uninfected control animals. We found that infection with both WT H18N11 and ΔNS1 resulted in a pronounced downregulation of genes belonging to metabolic pathways (oxidative phosphorylation, adipogenesis, fatty acid metabolism, and glycolysis) in the intestine (Fig. 2a and Supplementary Data 3). Interestingly, repression of genes associated with oxidative phosphorylation has also been observed in cells from IAV-infected mice[66]. In contrast, the same metabolic pathways remained either unchanged (fatty acid metabolism and glycolysis) or were inconsistently regulated (oxidative phosphorylation and adipogenesis) in the mesentery of infected bats compared with uninfected controls (Fig. 2b and Supplementary Data 3). Because most of the cell types in the intestine were epithelial and other non-immune cells, whereas the vast majority of the mesenteric cell types were immune cells (Fig. 1b, d and Supplementary Data 2), it is possible that these differences in the regulation of the metabolic pathways are due to distinct transcriptional responses of the cell types to H18N11 infection.

We further noticed a high enrichment of genes associated with the type I (α) and type II (γ) interferon responses, which was only observed at 3 but not at 9 dpi (Fig. 2a, b and Supplementary Data 3). To evaluate the innate immune response to infection in more detail, we performed a comparative gene expression analysis of genes encoding the following: interferons and pro-inflammatory cytokines (*Ifng*, *Ifnk*, *Il1b*, *Il6*, *Il18*, and *Tnf*), transcription factors involved in immunity and inflammation (*Irf1*, *Irf3*, *Irf7*, *Nfkb1*, and *Stat1*), interferon-stimulated genes (ISGs) (*Ifi6*, *Ifi27l1*, *Ifitm1*, *Isg15*, *Isg20*, *Ly6e*, *Mx1*, *Mx2*, *Oas1*, *Oas2*, *Oas3*, and *Oasl*), and pattern recognition receptors or nucleic acid sensors (*Ifih1*, *Nlrp3*, and *Sting1*). We found no significant induction of interferons or pro-inflammatory cytokines in most cell types (Fig. 2c, d). A very minor increase of *Il6* or *Tnf* transcripts was observed in intestinal enteric glial cells and mesenteric neutrophils of ΔNS1-infected bats and in intestinal ILC3s from WT-infected animals (Fig. 2c, d and Supplementary Fig. 7). However, as transient expression of innate-related genes in Jamaican fruit bats has been reported[67], we cannot exclude the possibility that the induction of these genes occurred immediately after infection and had already waned by 3 dpi. Among the transcription factor genes, *Irf7* was slightly induced in intestinal enterocytes and *Cd14*+ macrophages at 3 dpi, whereas *Stat1* was modestly upregulated in most mesenteric cell types at 3 dpi, particularly *Cd14*+ macrophages and neutrophils. In mesenteric neutrophils, we also observed a slight

increase in *Irf1* and *Nfkb1* transcript levels. This overall weak induction of interferon-related transcription factors could indicate either a dampened antiviral response or the remnants of waning signaling. A number of antiviral ISGs were moderately upregulated at 3 dpi: *Ifi6*, *Ifitm1*, *Ly6e*, *Mx1*, and *Isg15* in intestinal enterocytes, B cells, and *Cd14*+ macrophages, and *Ifi6*, *Ly6e*, *Mx1*, and *Isg15* in mesenteric B cells, *Cd14*+ macrophages, macrophages, and monocytes (Fig. 2c, d and Supplementary Fig. 7). Of these ISGs, *Ly6e* and *Ifi6* were the most strongly induced factors in the cells of both WT H18N11 and ΔNS1-infected bats. Importantly, other than mesenteric *Cd14*+ macrophages and neutrophils, none of the genes examined was more highly induced in the other cell types of ΔNS1-infected bats, indicating that WT and ΔNS1 infection elicited similar innate immune responses.

In conclusion, innate antiviral gene signatures were detected in both myeloid and lymphoid cell types, particularly *Cd14*+ macrophages, macrophages, and B cells, and were more pronounced at 3 dpi than at 9 dpi.

### Intestinal and mesenteric immune cells exhibit gene signatures associated with activation and effector function after infection

To examine the activation and effector function of immune cells in response to infection, we compared the gene expression levels of the immune cell types from WT and ΔNS1-infected animals with uninfected controls. In the intestine, we observed broad transcriptional responsiveness of almost all myeloid and lymphoid cell types, albeit to varying degrees (Fig. 3a). Specifically, mast cells exhibited increased expression of genes associated with mast cell development and motility, including *Gata1* and *Elmo1*[68,69]. In mast cells from ΔNS1-infected bats, we also observed enhanced levels of the activation markers *Lamp1* and *Il13*[70,71], which are induced in response to immunological stimulation. In addition, upregulation of the B-cell receptor signaling components *Ptprcap* and *Cd79a* indicated activation of intestinal plasma cells, which can express a functional B-cell receptor on the surface[72]. We also observed evidence for activation of ILC3s, characterized by induction of the gene encoding the tissue regenerative cytokine *Il22*[73], while ILC3s from ΔNS1-infected bats also exhibited moderate expression of transcripts for the pro-inflammatory cytokines *Il17a* and *Il17f* at 3 dpi[74,75]. In contrast, mesenteric immune cell populations were less responsive, with the notable exception of neutrophils (Fig. 3b). Mesenteric neutrophils exhibited upregulation of genes associated with tissue recruitment and motility, including the chemokine receptor *Cxcr2*, integrin beta-2 (*Itgb2*), and the transcription factor *Ier2* (Fig. 3b)[76]. Neutrophils from H18N11 ΔNS1-infected bats additionally showed induction of genes encoding the cell adhesion molecules *Cd44* and *Cd68*.

### Viral transcripts are found in intestinal epithelial cells and leukocytes

To identify the cell types infected by H18N11, we next screened the transcriptome samples for polyadenylated H18N11 mRNA transcripts in sequenced cells from the intestine and mesentery of bats infected with both WT H18N11 and ΔNS1. A total of 15 cells from the intestinal preparations collected at 3 dpi were found positive for at least one viral mRNA: 9 of 8979 cells (0.1%) from the WT H18N11-infected group and 6 of 5253 cells (0.11%) from the ΔNS1-infected group. These infected cells included enterocytes (*n* = 4), goblet cells (*n* = 1), LECs (*n* = 1), macrophages (*n* = 4), *Cd14*+ macrophages (*n* = 2), and B cells (*n* = 3) (Fig. 4a). Viral transcripts were also found in a total of 26 cells from the mesentery samples collected at 3 dpi: 19 of 2,315 cells (0.82%) from WT H18N11-infected bats and 7 of 3414 cells (0.2%) from ΔNS1-infected bats. Infected cells included monocytes (*n* = 9), macrophages (*n* = 3), *Cd14*+ macrophages (*n* = 2), cDC1s (*n* = 1), IDCs (*n* = 2), B cells (*n* = 1), and NK/T cells (*n* = 8) (Fig. 4b). No viral transcripts were found at 9 dpi. None of the infected cells harbored detectable levels of all eight viral mRNA segments; rather, in most virus transcript-positive cells, only

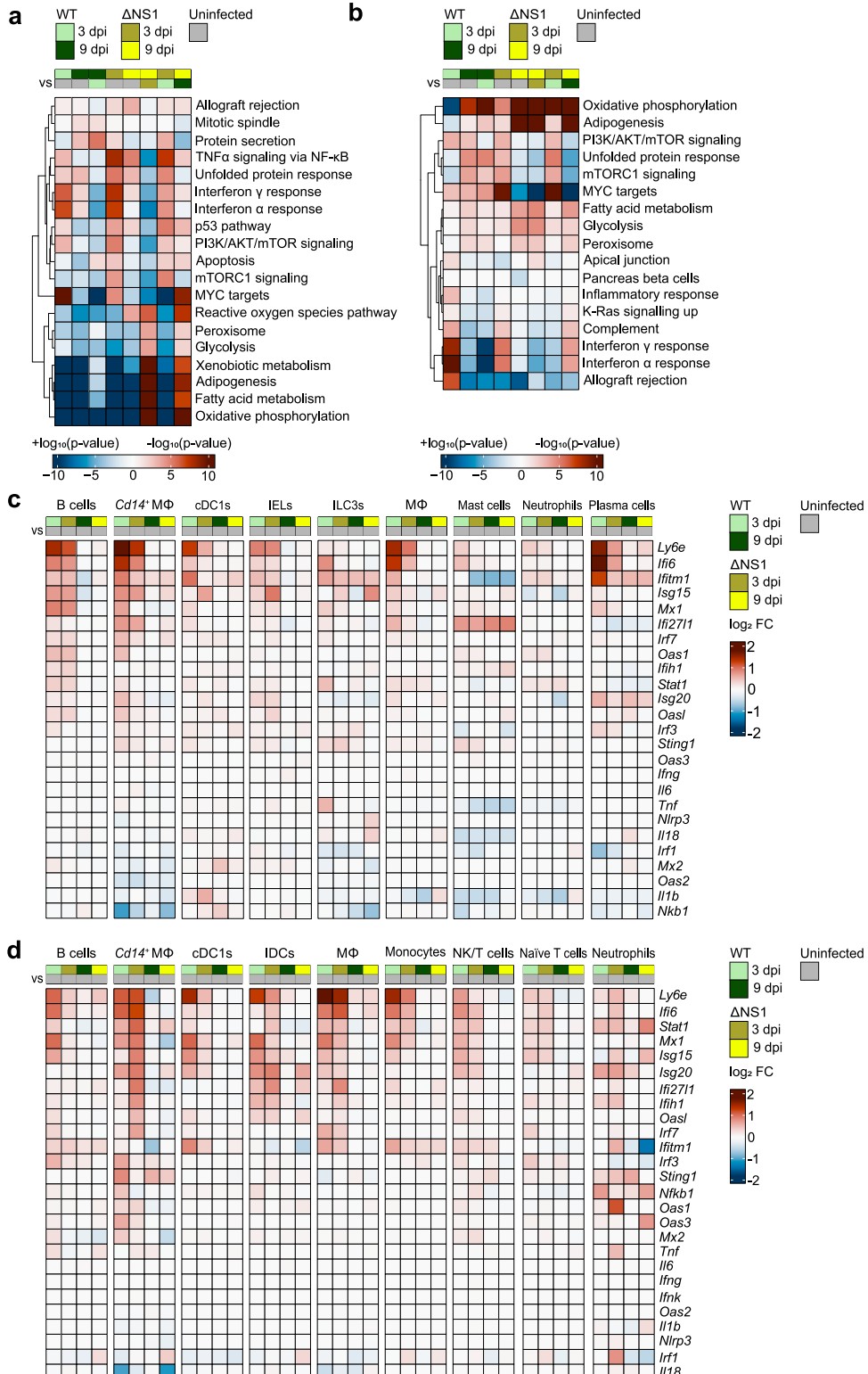

**Fig. 2 | Infection induces moderate upregulation of ISGs. a, b** Hallmark gene-set analysis of intestinal (**a**) and mesenteric cells (**b**) from WT H18N11 and ΔNS1-infected bats and uninfected controls. Enrichment is shown as −log$_{10}$ $p$ values for upregulated and the +log$_{10}$ $p$ values for downregulated gene sets, resulting in >0 or <0 values for up- and down-regulated gene sets, respectively. Statistical analysis was performed using a one-sided Fisher's exact test with a 'greater' alternative. **c, d** Gene-level heat maps showing log$_2$ fold changes relative to uninfected controls

in interferons and pro-inflammatory cytokines (*Ifng*, *Ifnk*, *Il1b*, *Il6*, *Il18*, and *Tnf*), transcription factors involved in immunity and inflammation (*Irf1*, *Irf3*, *Irf7*, *Nfkb1*, and *Stat1*), ISGs (*Ifi6*, *Ifi27l1*, *Ifitm1*, *Isg15*, *Isg20*, *Ly6e*, *Mx1*, *Mx2*, *Oas1*, *Oas2*, *Oas3*, and *Oasl*), and genes encoding pattern recognition receptors or nucleic acid sensors (*Ifih1*, *Nlrp3*, and *Sting1*) in immune cell types of the intestine (**c**) and mesentery (**d**). Source data are provided as a Source Data file.

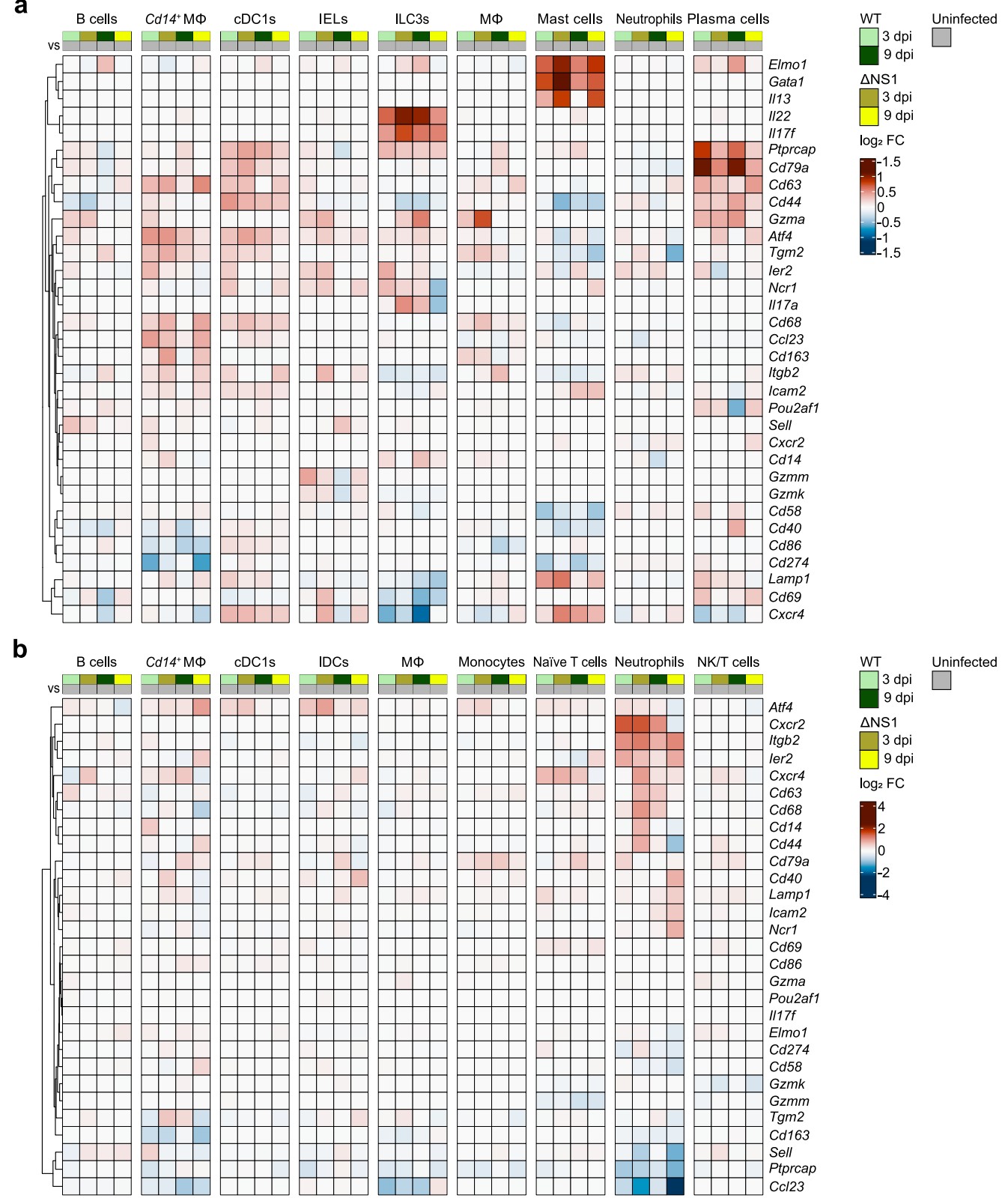

**Fig. 3 | Transcriptional activation and induction of effector genes in bat immune cells after viral infection. a**, **b** Gene-level heat maps showing log₂ fold changes in genes involved in activation and effector function in the indicated immune cells of the intestine (**a**) and mesentery (**b**) relative to uninfected controls. Hierarchical clustering on the Euclidean distance was used to cluster the genes according to their log₂ fold-change profiles. Source data are provided as a Source Data file.

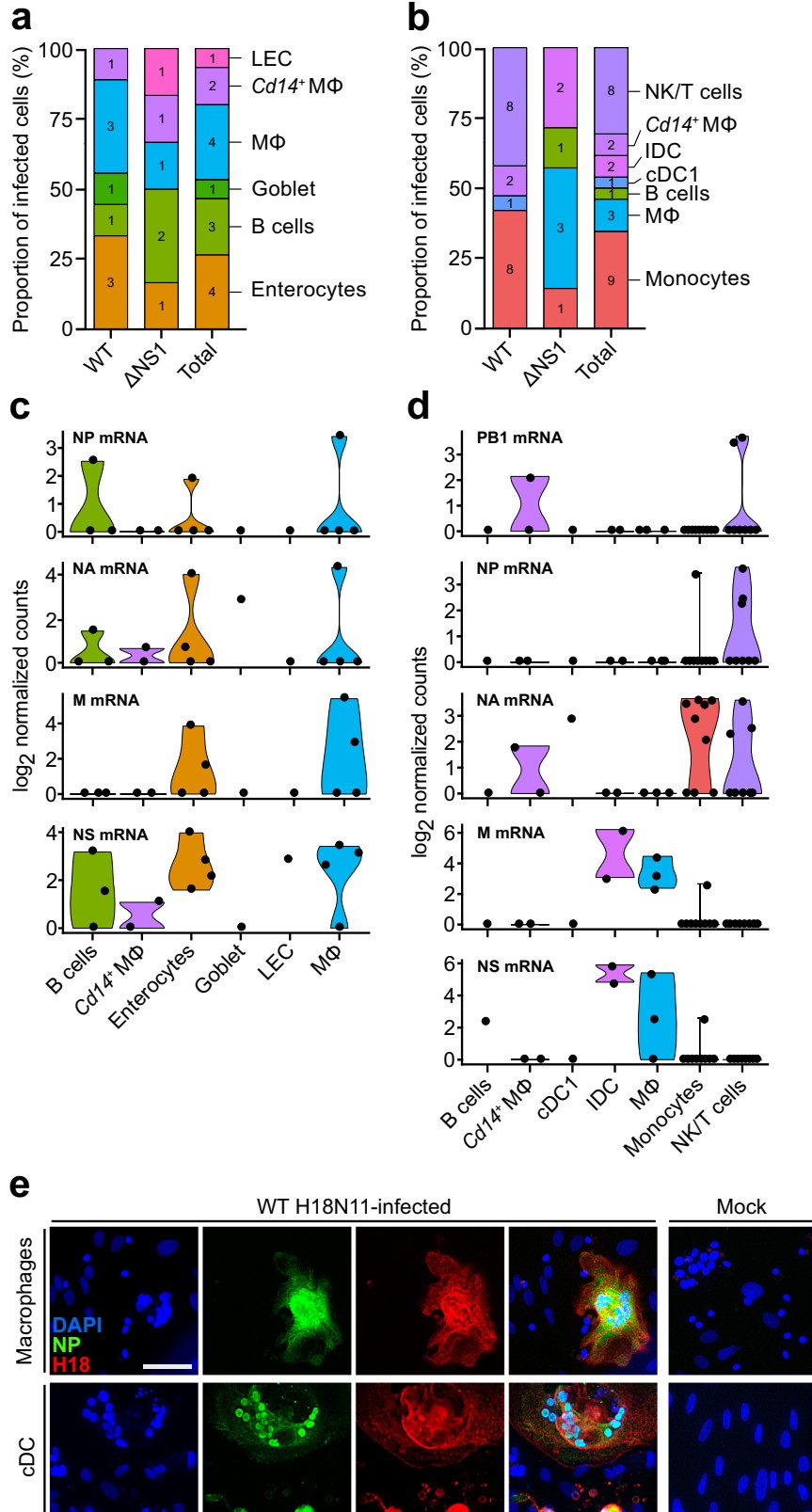

**Fig. 4 | H18N11 infection is predominantly observed in leukocytes. a, b** Bar graph showing the fraction and cell types infected by the WT H18N11 and ΔNS1 virus in the intestine (**a**) and the mesentery (**b**) at 3 dpi. The number of infected cells is shown. **c, d** Violin plots showing the log$_2$ normalized expression levels of the indicated viral mRNAs relative to the total expression in infected cells of the intestine (**c**) and mesentery (**d**). **e** Bat bone marrow-derived macrophages and cDCs were infected with WT H18N11 at an MOI of 3. At 48 hpi, samples were fixed, permeabilized, and probed with NP- and H18-specific antibodies. Representative images from *n* = 3 independent experiments. Scale bar, 50 μm. Source data are provided as a Source Data file.

one or two mRNAs were detected at 3 dpi (Fig. 4c, d). However, because synthesis of the detected viral mRNAs necessarily requires ongoing viral replication[77], the remaining mRNAs were most likely below the detection limit. The viral mRNA transcripts identified were NP, NA, M, and/or NS, as well as PB1 in *Cd14*+ macrophages and mesenteric NK/T cells, corresponding to the previously reported expression hierarchy of IAV genes[78,79]. In line with the detection of viral mRNAs in intestinal and mesenteric macrophages and cDC1s by scRNA-seq in vivo (Fig. 4c, d), bone marrow-derived macrophages and cDCs from Jamaican fruit bats were also susceptible to WT H18N11 infection ex vivo (Fig. 4e).

As the cell surface MHC-II molecule is essential for H18N11 infection[22,23], we searched for MHC-II transcripts within the scRNA-seq cell clusters from uninfected animals. However, we were unable to detect transcripts encoding the MHC-II receptor in our transcriptome data, likely due to incomplete annotation of the Jamaican fruit bat transcriptome used[80]. Nevertheless, we did find highly enriched levels of key MHC-II transcription factors (*Rfx5*, *Rfxap*, *Rfxank*, and *Ciita*), as well as the MHC-II-associated invariant chain (*Cd74*) and the alpha chain of the non-classical MHC-II chaperone Aj-DO (*Aj-Doa*) in macrophages, *Cd14*+ macrophages, cDC1s, IDCs, and B-cell clusters (Supplementary Fig. 8). Lower levels of the same genes were also found in enterocytes, goblet cells and NK/T cells, suggesting that these H18N11-susceptible cell types express the viral entry receptor.

### Detection of viral genomes in bat intestinal macrophages

Histopathological analysis of the infected Jamaican fruit bat intestine showed regularly structured intestinal villi with moderate infiltration of leukocytes into the lamina propria mucosae (Fig. 5a). Immunophenotyping revealed cytokeratin immunoreactivity in the epithelia (Fig. 5b), a predominance of Iba1+ macrophages in the lamina propria mucosae (Fig. 5c), and few intraepithelial CD20+ B cells (Fig. 5d). Consistent with the relatively low number of viral mRNA positive cells in scRNA-seq, only isolated H18-vRNA signals were detected in the intestinal mucosa of infected bats by chromogenic RNAscope in situ hybridization (Fig. 5e). Using confocal microscopy, we identified H18-vRNA in Iba1+ macrophages (Fig. 5f). Abundant H18-vRNA spots were localized within the center of the macrophage, interpreted as the nucleus, and a smaller fraction co-localized with cytoplasmic Iba1 signals (Fig. 5g, h), suggesting active viral replication and virion formation.

### Human monocyte-derived macrophages are highly susceptible to H18N11 infection

Human MHC-II molecules also function as a receptor for H18-mediated cell entry[22], raising concerns that bat IAV may have a zoonotic potential[81]. To test for general susceptibility and compare the profile of susceptible cells to our findings from Jamaican fruit bats, we first inoculated human peripheral blood mononuclear cells (PBMCs) with a high multiplicity of infection (MOI of 5) of WT H18N11 and determined the number of infected cells based on H18 expression. We found that 0.13% of total PBMCs were infected at 24 hpi, including 0.5% of myeloid cells (CD11b+ population), 2% of B cells (CD19+ population), and 0.16% of T cells (CD3+ population; Fig. 6a–c). In addition, infection of human PBMCs was not productive, as we did not detect an increase in newly generated infectious viral particles (Fig. 6d). Based on the infection of macrophages in bats, we inoculated human monocyte-derived macrophages with WT H18N11 or ΔNS1 (Fig. 6e, f). At 24 hpi, 67 and 42% of the macrophages inoculated with WT H18N11 and ΔNS1 were infected, resulting in 40% and 8% dead cells, respectively (Fig. 6f, g). Monocyte-derived macrophages also supported robust multicycle viral replication of the WT virus in a dose-dependent manner, resulting in peak titers of $8.9 \times 10^4$ and $3.7 \times 10^6$ FFU mL$^{-1}$ between 60 and 72 hpi (Fig. 6h, i). However, high-dose infection of macrophages with ΔNS1 resulted in more than 100-fold lower replication compared to WT

H18N11 (Fig. 6i), highlighting the importance of NS1 as a virulence factor for efficient viral replication in human leukocytes. Collectively, these results demonstrate that H18N11 can infect and replicate in human leukocytes, with myeloid cells, particularly macrophages, being significantly more susceptible than lymphoid cells.

## Discussion

In this study, we generated single-cell transcriptome atlases of the Jamaican fruit bat intestine and mesentery and investigated the response to infection with the bat-derived IAV H18N11. This combination of virus and host represents a unique opportunity to study a bat-derived pathogen with zoonotic potential in its natural host under controlled laboratory conditions. Furthermore, the gastrointestinal tract appears to be a crucial site of viral replication and transmission for H18N11 as well as other bat viruses of zoonotic concern[25,82–85]. Our work provides an in-depth characterization of the Jamaican fruit bat immune cell composition and dynamics in these organs.

Our analysis shows that the bat intestinal and mesenteric environment is largely populated by canonical immune cell types, including seven myeloid and six lymphoid subsets. A notable exception among the lymphoid cells were three indistinguishable NK/T cell clusters that exhibited characteristics of both NK and T cells, including expression of genes encoding activating (*Klrd1* and *Klrk1*) and inhibitory (*Klrd1* and *Klrg1*) KLRs, as well as activating NCRs (*Ncr1* and *Ncr3*), TCRs, and their co-receptors. NK/T cells were the predominant lymphoid cell type in the mesentery of uninfected Jamaican fruit bats and expanded immediately after infection with H18N11. Surprisingly, NK/T cells showed only little transcriptional activation and no induction of effector genes after infection. Recent studies have also identified NK/T cells in the distantly related Egyptian rousette bat and cave nectar bat[31,32], suggesting that they are a general component of the bat immune cell repertoire. Although NK/T cells were found to be the dominant immune cell in the lungs of cave nectar bats[32], and have undergone receptor expansion and diversification in the Egyptian rousette bats[63], more research is required to understand the bat-specific evolution of NK/T cell functions. The same is true for the role of bat B cells in the context of a virus-induced immune response. Previous studies have shown that bat B cells are functional[86], but the function of the antibodies they produce, as well as the magnitude and duration of virus-induced antibody responses, remain largely unclear[87]. Our data suggest that B cells may be critical for the antiviral response in Jamaican fruit bats, because they rapidly became the dominant lymphoid immune cell type in the intestine and mesentery after infection, and also underwent antigen-specific activation and maturation, as indicated by the expression of *Aicda* and detectable antibody titers. Therefore, adaptive immune activation of the Jamaican fruit bat may contribute to the control and clearance of H18N11 infection.

Although scRNA-seq is a powerful tool to identify tissue-specific cell repertoires and their infection-related changes, the as-yet incomplete annotation of the Jamaican fruit bat genome and our usage of human and mouse-specific cell marker genes represents a critical limitation to resolving the precise nature and function of the bat immune cell types. Because immune markers are conserved between humans and bats, we streamlined our workflow in this study by first aligning our reads to the available Jamaican fruit bat transcriptome and then mapping them to human orthologs[19,88]. In the future, an improved Jamaican fruit bat transcriptome and further functional protein-based studies are needed to define bat-specific lineage markers in order to understand the full extent of the bat immune response to infection, and to facilitate comparative immunology studies.

By combining our scRNA-seq findings with spatially resolved in situ hybridization, we show that immune cells, particularly macrophages, are the primary targets of the bat-derived IAV H18N11. As expected from our previous and current histopathological findings

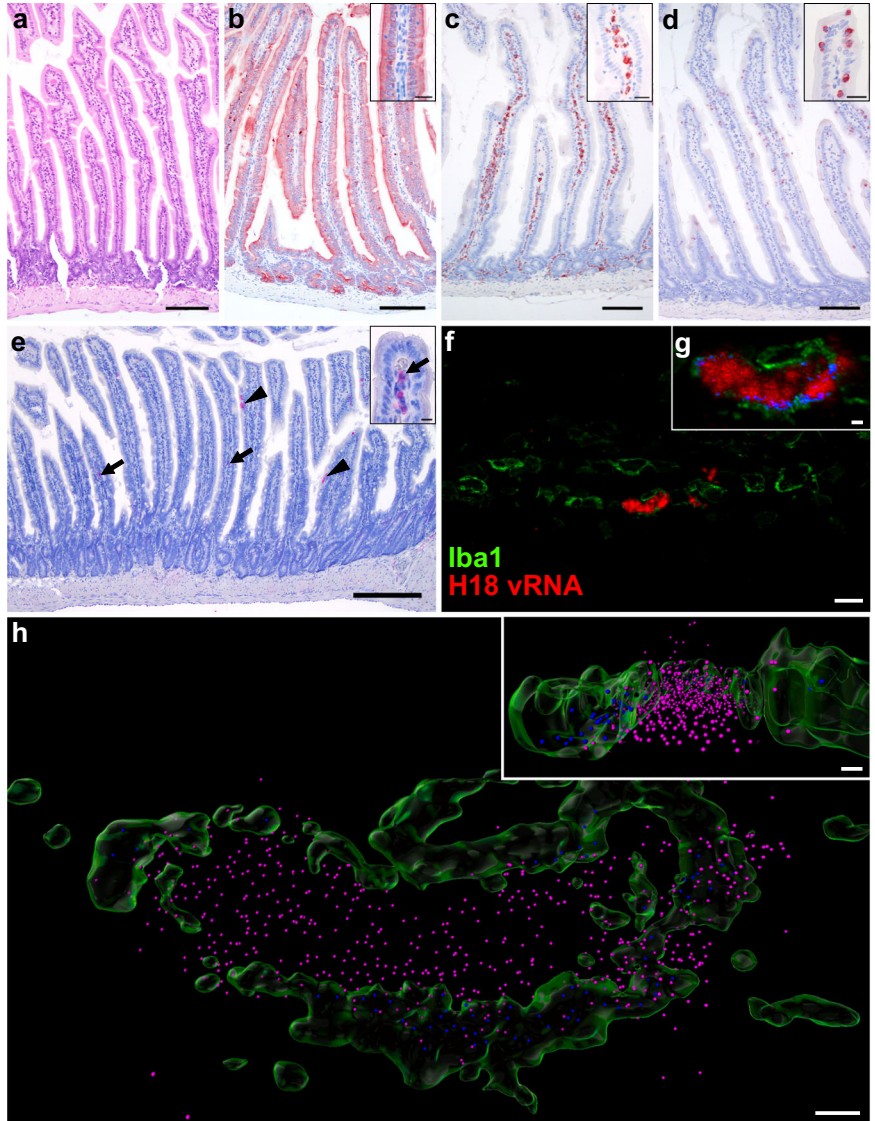

**Fig. 5 | H18N11 replicates in Iba1+ macrophages in the bat intestine.**
**a**–**e** Histology and immunohistochemistry of the small intestine of H18N11-infected Jamaican fruit bats (*n* = 3). **a** Hematoxylin & eosin-stained small intestine. Scale bars, 100 μm (main panel). **b** Wide-spectrum cytokeratin+ enterocytes and crypt epithelia (red), with strong antigen expression in the apical cell regions. Scale bars, 100 μm (main panel), 10 μm (inset). **c** Many Iba1+ macrophages (red) within the lamina propria mucosae. Scale bars, 100 μm (main panel), 10 μm (inset). **d** Few CD20+ B-cells (red) within the mucosal epithelium. Scale bars, 100 μm (main panel), 10 μm (inset). **e** Transversal section of the small intestine single H18-vRNA signals (red) in the lamina propria mucosae (arrows) and in epithelia (arrowheads). The inset shows multiple discrete H18-vRNA signals in association with mononuclear cells. Scale bars, 200 μm (main panel), 10 μm (inset). **f**–**h** Confocal laser scanning microscopy of the lamina propria mucosae of H18N11-infected Jamaican fruit bats (*n* = 2). **f** Iba1+ macrophages (green) and H18-vRNA (red). Scale bar, 10 μm. **g** Co-localization (blue) of Iba1 (green) and H18-vRNA signals (red). Scale bar, 1 μm. **h** Transparent surface rendering shows cytoplasmic Iba1 signals (green) and distinct H18-vRNA spots in the cell center (magenta) or H18-vRNA spots co-localizing with cytoplasmic Iba1 (blue). The inset shows a transversal section through the infected macrophage. Scale bars, 1 μm (main panel), 0.5 μm (inset).

showing confined oligofocal H18N11 infection in bat intestine and lymph nodes[25], we detected a limited number of virus-positive cells with low viral load by scRNA-seq. In contrast, mice infected with classical IAVs exhibited both higher fractions of infected cells and higher viral loads[66,89], suggesting that cellular infection dynamics may differ between hosts (bat vs. mouse) and viruses (conventional IAV vs. bat-derived IAV). Moreover, droplet-based scRNA-seq approaches, such as the one used here, have relatively low sensitivity[90,91] and likely fail to detect all viral transcripts. This is supported by our observation that high levels of HA genome segments, and thus also HA mRNAs, were detected in tissue-resident macrophages by in situ hybridization but not by scRNA-seq.

Although the exact sequence of events leading to infection of intestinal and mesenteric leukocytes is unclear, we propose the following hypothesis: H18N11 is ingested through fecal-oral transmission and traverses the digestive tract, where intestinal resident macrophages and DCs that sample the intestinal lumen via transepithelial dendrites actively ingest the virus or become infected. Whether enterocytes become infected at this stage remains unclear. In the next step, infected immune cells migrate to the mesenteric lymph node and spread the infection to other leukocytes. This hypothesis is supported by our scRNA-seq data, where comparable numbers of virus-positive cells were detected in the intestine for WT H18N11 (9 cells) and the NS1-deficient variant (6 cells), but fewer infected cells in the mesentery of ΔNS1-infected bats (7 cells) compared to WT (19 cells), suggesting that the ΔNS1 virus was cleared before robust infection of mesenteric leukocytes could be established. Our proposed H18N11 infection cycle would be similar to that of the measles virus, in which it initially infects

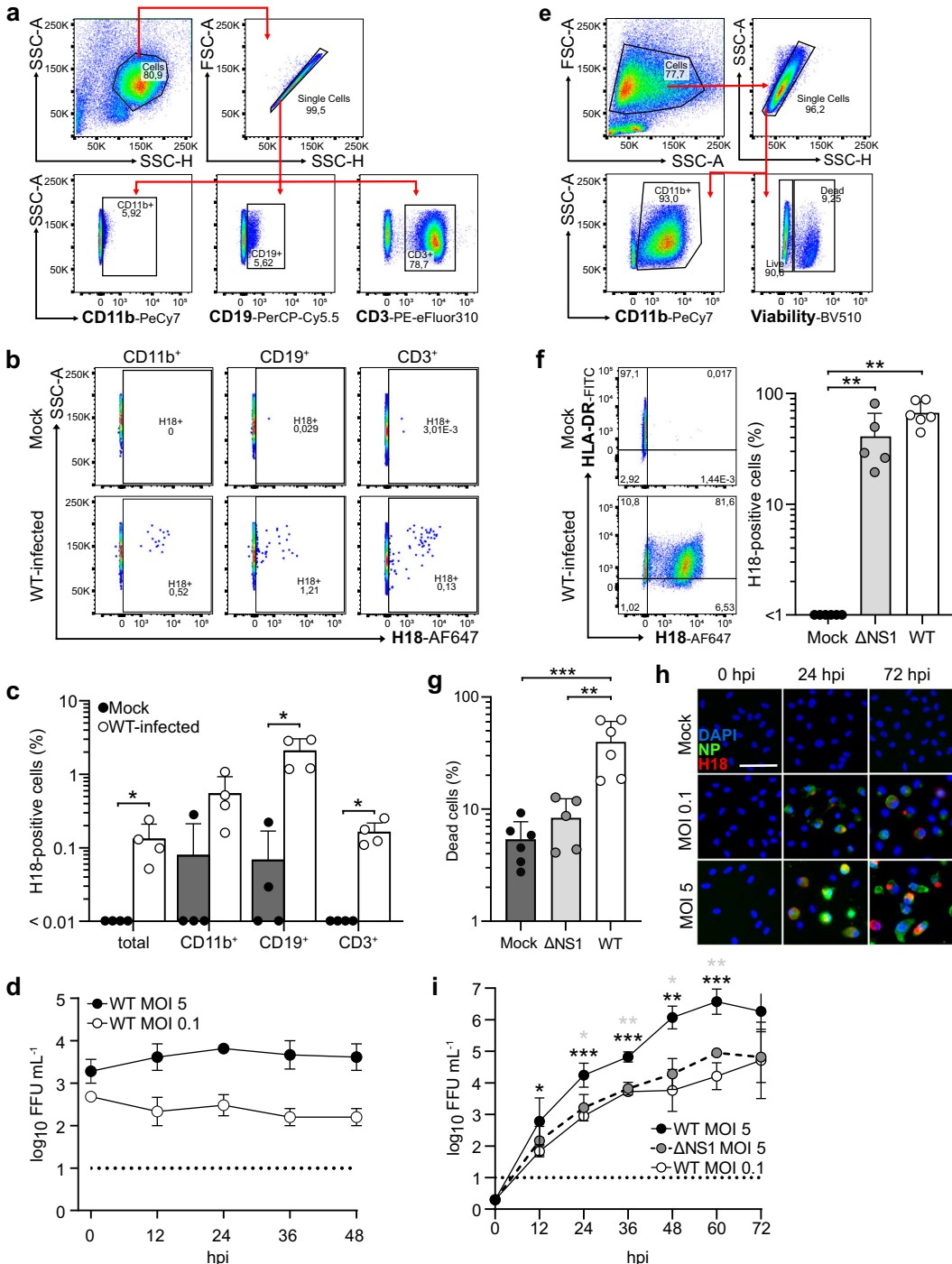

alveolar macrophages and dendritic cells, which then ferries it to draining lymph nodes to infect B and T lymphocytes[92,93]. However, more detailed analyses are required to further support this hypothesis.

Our study also highlights the importance of NS1 for efficient viral replication, as deletion of the IAV innate antagonist NS1 impaired the virus' ability to replicate efficiently in Jamaican fruit bats. Interestingly, infection with ΔNS1 resulted in similar transcriptional changes compared to infection with the WT H18N11. Further experiments are needed to determine whether the deletion of NS1 affected the viral immunogenicity or whether the immune control mechanisms of the Jamaican fruit bat prevented an overly strong immune response. In addition, NS1 has been shown to be a multifunctional protein in classical IAVs that interacts with other host cell proteins to regulate apoptosis and promote viral gene expression in addition to its

immunomodulatory properties[33,94]. Hence, it is conceivable that truncation of the H18N11 NS1 may have impaired such functions that negatively affected efficient viral replication.

Consistent with the finding that H18N11 can use human MHC-II molecules for cell entry[22,23], we show that H18N11 is also capable of infecting several human leukocyte subpopulations at low levels, but replicates efficiently in human macrophages. Considering that MHC-II molecules from all species tested to date facilitate H18-mediated viral entry[22], robust replication in human macrophages suggests that the host range may not necessarily be limited to bats and argues for further studies to fully assess the zoonotic risk of this bat IAV. Therefore, further surveillance and sero-epidemiological studies in non-bat species and humans living within the H18N11 distribution range in Central and South America are warranted to assess the exposure to H18N11.

**Fig. 6 | H18N11 infects human B and T cells and replicates in macrophages.**
**a** Flow cytometry gating strategy for human PBMCs. **b** Flow cytometry to identify intracellular H18 expression in H18N11-infected myeloid cells (CD11b⁺), B cells (CD19⁺), and T cells (CD3⁺) among human PBMC at 24 hpi at an MOI of 5. Representative images from $n = 4$ independent experiments. **c** Bar graphs showing quantification of WT H18N11-infected PBMC subsets compared to mock controls. Data are mean ± SD of $n = 4$ independent experiments. Statistical analysis was performed using a two-tailed Mann–Whitney test (*$p = 0.0286$). **d** Viral growth kinetics in PBMCs infected with WT H18N11 at the indicated MOI. Viral titers were determined by endpoint titration at the indicated time points. Viral titers are the mean ± SD of $n = 3$ independent experiments. The dashed line indicates the detection limit. **e** Flow cytometry gating strategy for human monocyte-derived macrophages. **f** Flow cytometry to detect intracellular H18 expression in monocyte-derived macrophages infected with WT H18N11 or ΔNS1 at an MOI of 5 at 24 hpi. Representative images from $n \geq 5$ independent experiments. The bar graph shows the quantification of H18-positive cells. Data are mean ± SD of $n = 5$ (ΔNS1) or $n = 6$ (mock, WT H18N11) independent experiments. Statistical analysis was performed using a two-tailed Mann–Whitney test (**$p = 0.0022$ for mock vs. WT H18N11 and

**$p = 0.0043$ for mock vs. ΔNS1). **g** Flow cytometry to determine the viability of monocyte-derived macrophages infected with WT H18N11 or ΔNS1 at an MOI of 5 at 24 hpi. The bar graph shows the quantification of dead cells. Data are mean ± SD of $n = 5$ (ΔNS1) or $n = 6$ (mock, WT H18N11) independent experiments. Statistical analysis was performed using a Tukey's multiple comparison test with single pooled variance (***$p = 0.0008$ for mock vs. WT H18N11, **$p = 0.0025$ for WT H18N11 vs. ΔNS1). **h** Subcellular localization of viral NP (green) and H18 (red) antigens in WT H18N11-infected monocyte-derived macrophages was monitored over time by immunofluorescence. Representative images from $n = 3$ independent experiments. Scale bar, 50 μm. **i** Viral growth kinetics in monocyte-derived macrophages infected with WT H18N11 or ΔNS1 at the indicated MOI. Viral titers were determined by endpoint titration at the indicated time points. Viral titers are the mean ± SD of $n = 4$ independent experiments; statistical analysis was performed using Dunnett's multiple comparison test. *$p < 0.05$, **$p < 0.01$, ***$p < 0.001$; ****$p < 0.0001$. Black asterisks, WT H18N11 MOI 5 vs. MOI 0.1; gray asterisks WT H18N11 MOI 5 vs. ΔNS1 MOI 5. The dashed line indicates the detection limit. Source data are provided as a Source Data file.

---

Taken together, our results depict a cellular landscape in the Jamaican fruit bat intestine and mesentery comprised of numerous canonical epithelial, endothelial, and immune cells together with several non-canonical immune cell populations. The relatively mild transcriptional antiviral response to H18N11 infection along with the lack of an excessive response to ΔNS1 provides several entry points to further examine the apparent relative tolerance of bats to H18N11 and possibly even other viral infections. Our findings of H18N11 tropism for several leukocyte populations in bats and humans also point to potential infection routes in other species.

## Methods

### Ethics statement
Our research complies with all relevant ethical regulations. The bat experiments were approved by the Colorado State University Institutional Animal Care and Use Committee (IACUC) protocol #1574. Written informed consent was obtained from all participants for the isolation of human peripheral blood mononuclear cells (PBMCs), and the study was conducted in accordance with federal guidelines, local ethics committee regulations (Ethik-Kommission Universitätsklinikum Freiburg, reference number 20-1109), and the Declaration of Helsinki (1975).

### Cell lines
HEK-293T cells were obtained from the American Type Culture Collection (ATCC; CRL-3216). MDCK II cells stably overexpressing the human MHC-II molecule HLA-DR15 (MDCK II-MHC-II) were generated previously[22].

### Generation of recombinant bat influenza H18N11 viruses
The pHW2000-based rescue system of A/flat-faced bat/Peru/033/2010 (H18N11)[25] was used to generate recombinant wild-type H18N11 and the NS1-deficient variant H18N11, designated ΔNS1, which has a C-terminal deletion in NS1 resulting in a protein of 128 amino acids in length. The corresponding pHW2000 plasmids encoding the truncated ΔNS1 segment or the NS reporter segment were generated by Gibson assembly. Briefly, HEK-293T cells were transfected with 300 ng of each of the eight pHW2000 plasmids encoding PB2, PB1, PA, HA, NP, NA, M, and NS or ΔNS1 segment of A/flat-faced bat/Peru/033/2010 (H18N11). In addition, 400 ng of each of the four pCAGGs expression plasmids encoding PB2, PB1, PA, and NP of H18N11 were co-transfected. At 48 h post-transfection, the virus-containing supernatant was collected and transferred in fresh infection medium (Dulbecco's modified Eagle medium (DMEM, Gibco) containing 0.2% bovine serum albumin (BSA, Sigma-Aldrich), 100 U mL⁻¹ penicillin (Sigma-Aldrich), 100 μg mL⁻¹ streptomycin (Sigma-Aldrich) and N-tosyl-L-phenylalanine

chloromethyl ketone (TPCK)-treated trypsin (Sigma-Aldrich)) to MDCK II-MHC-II pretreated with 1 mg mL⁻¹ of DEAE-dextran (Sigma-Aldrich) in phosphate-buffered saline (PBS) containing 0.2% BSA (Sigma-Aldrich) 1 h prior to infection. The supernatant was collected 72 h after infection and concentrated by ultracentrifugation through a 30% sucrose cushion.

### Jamaican fruit bat in vivo infection
All work with bats was approved by the Colorado State University (CSU) Institutional Animal Care and Use Committee, protocol 1574. Jamaican fruit bats were obtained from the breeding colony by the Schountz laboratory housed at CSU. This colony has been closed for 17 years and has been determined to be free of influenza virus infection by serology (ELISA) and lack of detection of vRNA (rectal swabs). Bats were housed in separate rooms in bird cages for the duration of the experiments. Jamaican fruit bat infection experiments were performed at the Center for Vector-borne Infectious Diseases at CSU under animal biosafety level 2 (ABSL-2) containment with approvals from the CSU Institutional Biosafety Committee and Animal Care and Use Committee. All investigators and animal caretakers followed BSL-2 biosafety and infection control practices and wore personal protective equipment, including PAPRs. Because pregnancy could affect the bats' immune systems and virus replication, only male bats with an age of 1 to 5 years were used in the experiments. For inoculation, Jamaican fruit bats (*Artibeus jamaicensis*) were anesthetized with 5% isoflurane in $O_2$ and then oronasally inoculated with 12.5 μL per nostril of either WT ($n = 6$) or ΔNS1 ($n = 6$) ($5 \times 10^5$ TCID$_{50}$ in 25 μL). Sham-infected bats ($n = 4$) were included as controls. To determine viral shedding, bat rectal swab material was collected at 2 and 7 dpi in 500 μL brain heart infusion broth (BHI) (Thermo Fisher Scientific). At 3 and 9 dpi WT- and ΔNS1-infected ($n = 3$ per group and time point) and uninfected control bats ($n = 2$ at each time point) were euthanized by inhalation of isoflurane followed by thoracotomy. Organs and blood were collected at necropsy and processed for scRNA-seq and serology.

### Isolation and in vitro infection of bat bone marrow-derived cDCs and macrophages
Bat bone marrow was isolated from the humeri and ulnas by rinsing the inside of the bones with serum-free HBSS (Corning). Collected bone marrow was centrifuged at $350 \times g$ at room temperature for 10 min and washed with Roswell Park Memorial Institute (RPMI)−1640 medium (Corning) containing 2% fetal calf serum (FCS), 10 μg mL⁻¹ gentamicin (Sigma-Aldrich), 100 U mL⁻¹ penicillin (Sigma-Aldrich), 100 μg mL⁻¹ streptomycin (Sigma-Aldrich), 200 mM L-glutamine (Cytiva), 10 mM HEPES (Teknova), and 40 μM 2-mercaptoethanol (Gibco). Bone marrow cells were seeded in 100 mm Petri dishes at a density of $2 \times 10^6$

cells in macrophage medium (RPMI-1640 medium containing 10% FCS, 10 µg mL$^{-1}$ gentamicin, 100 U mL$^{-1}$ penicillin (Sigma-Aldrich), 100 µg mL$^{-1}$ streptomycin (Sigma-Aldrich), 200 mM L-glutamine (Cytiva), 10 mM HEPES (Teknova), 40 µM 2-mercaptoethanol (Gibco)) containing 20 ng ml$^{-1}$ Egyptian fruit bat macrophage colony-stimulating factor (M-CSF; Kingfisher Biotech, Inc.) to generate macrophages, or 20 ng mL$^{-1}$ recombinant Jamaican fruit bat granulocyte-macrophage colony-stimulating factor (GM-CSF; Kingfisher Biotech, Inc.) to generate common dendritic cells (cDC). Cells were incubated for 10 to 14 days at 37 °C and 5% CO$_2$ with exchange of medium every two days. Bat cDCs and macrophages were inoculated with WT H18N11 at an MOI of 5 for 3 h in infection medium (RPMI containing 0.2% BSA (Sigma-Aldrich) and 100 U mL$^{-1}$ penicillin (Sigma-Aldrich) and 100 µg mL$^{-1}$ streptomycin (Sigma-Aldrich) and 1 µg mL$^{-1}$ TPCK-treated trypsin) and were subsequently washed with PBS and cultivated in fresh macrophage medium. At 24 hpi, cells were fixed with 4% paraformaldehyde (PFA, EMD) and permeabilized with 0.5% Triton X-100 (Sigma-Aldrich). Viral antigens were detected with polyclonal rabbit anti-H18 (in-house made, 1:500) and monoclonal mouse anti-NP (in-house made, 1:500) in PBS containing 5% normal goat serum (NGS, Vector Laboratories, Inc.). Primary antibodies were detected using secondary goat anti-rabbit (Jackson ImmunoResearch, 1:500) conjugated to Cy3 and goat anti-mouse conjugated to AlexaFluor488 (Jackson ImmunoResearch, 1:500) in PBS containing 5% NGS (Vector Laboratories, Inc.). Nuclei were visualized using DAPI (Invitrogen).

### Isolation and in vitro infection of human peripheral blood mononuclear cells (PBMCs) and monocyte-derived macrophages

Human peripheral blood mononuclear cells (PBMCs) were isolated from the venous blood of three healthy female donors with a median age of 28 years (Ethik-Kommission Universitätsklinikum Freiburg, reference number 20-1109) by density gradient centrifugation with a Ficoll® Paque Plus (Cytiva) gradient. Freshly isolated PBMCs were infected with WT H18N11 or ΔNS1 in infection medium (RPMI containing 0.2% BSA (Sigma-Aldrich), 100 U mL$^{-1}$ penicillin (Sigma-Aldrich) and 100 µg mL$^{-1}$ streptomycin (Sigma-Aldrich) and 1 µg mL$^{-1}$ TPCK-treated trypsin) at the indicated MOI by spin-infection for 90 min at 1600×g and 37 °C. Subsequently, the inoculum was removed and replaced by fresh infection medium and cells were cultured at 37 °C and 5% CO$_2$. To determine the susceptibility of PBMCs to H18N11 infection, cells were harvested at 24 hpi and stained for flow analysis. Live-dead-staining was performed with Zombie Violet™ fixable viability dye (BioLegend). Human Fc-blocking reagent (BD Pharmingen™) was used at a concentration of 0.5 µg µL$^{-1}$ in FACS buffer (PBS containing 1% BSA (Sigma-Aldrich), 0.05% sodium acid and 5 mM ethylenediaminetetraacetic acid (EDTA, Corning)) according to the manufacturer's instructions. Cell surface antigens were stained with BV605-conjugated mouse anti-human CD3 (BioLegend, 1:100), PerCP-Cy5.5-conjugated mouse anti-human CD19 (BioLegend, 1:100), PE/Cy7-conjugated mouse anti-human CD11b (Invitrogen, 1:400) and FITC-conjugated mouse anti-human HLA-DR (BioLegend, 1:100) in FACS buffer. Cells were then fixed and washed with the Cytofix/Cytoperm™ Fixation/Permeabilization kit (BD Biosciences), and viral antigens were stained using an Alexa Fluor™ 647-conjugated mouse anti-H18 (in-house made, conjugated with Alexa Fluor™ 647 Antibody Labeling Kit (Invitrogen) according to manufacturer's instructions, 1:500). Fluorescent-minus-one (FMO) controls were prepared with the respective antibodies spared out. Single stains were prepared with the AbC™ Total Antibody Compensation Bead Kit (Invitrogen) or the ArC™ Amine Reactive Compensation Bead Kit (Invitrogen) for the viability dye according to the manufacturer's instructions. Finally, cells were resuspended in FACS buffer and sieved into FACS tubes. Samples were analyzed using a BD LSRFortessa™ Cell Analyzer and FlowJo v10.8.1. software.

For multicycle growth kinetics, PBMCs were infected as described above at an MOI of 0.1 or 5, and the supernatant was collected at the indicated time points. Viral titers were determined by endpoint titration on MDCK II-MHC-II cells.

To obtain monocyte-derived macrophages, human PBMCs were resuspended in macrophage growth medium (RPMI containing 10% FCS, 100 U mL$^{-1}$ penicillin (Sigma-Aldrich), 100 µg mL$^{-1}$ streptomycin (Sigma-Aldrich), and 40 ng mL$^{-1}$ human macrophage colony-stimulating factor (hM-CSF, Prepotech)) and cultured at 37 °C and 5% CO$_2$. The next day, cells were washed once, and a fresh macrophage growth medium was added. Cells were cultured for 7 to 8 days with exchange of medium every 2 days. Monocyte-derived macrophages were infected with H18N11 in the infection medium at the indicated MOI for 1 h at 37 °C and 5% CO$_2$. Subsequently, the inoculum was removed and the cells were washed with PBS, followed by a pH shock with PBS pH 2 for 30 s and two more washing steps with PBS. Fresh infection medium was added and cells were incubated at 37 °C and 5% CO$_2$. Infection of macrophages with H18N11 was determined by immunofluorescence. Briefly, macrophages were fixed at the indicated time points with 4% PFA (EMD) and the cell membrane was permeablized using PBS containing 0.5% Triton X-100 (Sigma-Aldrich). Viral antigens were visualized using polyclonal rabbit anti-H18 (in-house made, 1:500) and mouse anti-NP (in-house made, 1:500) in PBS containing 5% NGS (Vector Laboratories, Inc.). Cells were washed with PBS and incubated with secondary goat anti-rabbit (Jackson ImmunoResearch, 1:500) conjugated to Cy3 and goat anti-mouse conjugated to AlexaFluor488 (Jackson ImmunoResearch, 1:500) in PBS containing 5% NGS (Vector Laboratories, Inc.). Nuclei were visualized using DAPI (Invitrogen). Fluorescent images were acquired using a Zeiss Observer.Z1 microscope with an AxioCam MRc camera.

Infection of monocyte-derived macrophages was also determined by flow cytometry. Macrophages were detached using accutase (Thermo Fisher Scientific) and mechanical treatment with a cell scraper. Cells were then washed with PBS containing 10% FCS, followed by two additional washing steps with PBS containing 5 mM EDTA (Corning). Life-Dead-Staining was performed with Zombie NIR™ fixable viability dye (BioLegend, 1:500) or Zombie Aqua™ fixable viability dye (BioLegend, 1:500). Human Fc-blocking reagent (BD Pharmingen™) was used at a concentration of 0.5 µg µL$^{-1}$ in FACS buffer according to the manufacturer's instructions. Cell surface antigens were stained with PE-conjugated mouse anti-human CD206 (Invitrogen, 1:100), PeCy7-conjugated mouse anti-human CD11b (Invitrogen, 1:400), and FITC-conjugated mouse anti-human HLA-DR (BioLegend, 1:100) in FACS buffer. Macrophages were then fixed and washed with the Cytofix/Cytoperm™ Fixation/Permeabilization kit (BD Biosciences), and viral antigens were stained as described for PBMCs. Samples were analyzed using a BD LSRFortessa™ Cell Analyzer and FlowJo v10.8.1. software.

For multicycle growth kinetics, monocyte-derived macrophages were infected as described above at an MOI of 0.1 or 5, and the supernatant was collected at the indicated time points. Viral titers were determined by endpoint titration on MDCK II-MHC-II cells.

### Virus titration

Viral titers were determined using immunofluorescence on subconfluent MDCK II-MHC-II cells pretreated with PBS containing 0.2% BSA (Sigma-Aldrich) and 1.5 mg mL$^{-1}$ DEAE-dextran. At 48 hpi, the inoculum was removed, cells were fixed with 4% PFA (EMD), and the cell membrane was permeabilized with PBS containing 0.5% Triton X-100 (Sigma-Aldrich). Infected cells were stained with a polyclonal rabbit anti-H18 (in-house made, 1:500) and mouse anti-NP (in-house made, 1:500) in PBS containing 5% NGS (Vector Laboratories, Inc.). Primary antibodies were detected using secondary goat anti-rabbit (Jackson ImmunoResearch, 1:500) conjugated to Cy3 and goat anti-mouse conjugated to AlexaFluor488 (Jackson ImmunoResearch, 1:500) in PBS containing 5% NGS (Vector Laboratories, Inc.).

## Preparation of single-cell suspension from mesentery and intestine of Jamaican fruit bats

After the incision of the right atrium, the cardiovascular system of each bat was perfused with 60 mL of perfusion solution consisting of 1×PBS, 5 mM EDTA (Corning), and 10 U mL⁻¹ heparin by injection into the left ventricle. The small and large intestines and mesentery were removed from the abdominal cavity and separated. To generate a single-cell suspension of the intestine, small and large intestines were removed and cut longitudinally to expose the lumen, intestines were then cut into 0.5 to 1.0 cm pieces and washed four times with 10 mL wash solution consisting of 1×PBS, 5 mM EDTA (Corning) under vigorous shaking. Subsequently, 20 mL of predigestion buffer (Hanks' balanced salt solution (HBSS) containing 20 mM HEPES (Teknova), 5 mM EDTA (Corning), 5% FCS, and 1 mM Dithiothreitol (DTT, Millipore-Sigma)) were added and incubated at 37 °C for 20 min with constant rotation. After vortexing for 10 s, the pieces were drained through a 70 μm cell strainer and the supernatant was saved. Cells in the supernatant were washed three times by adding wash solution and subsequent centrifugation at 350×$g$ for 5 min. The predigestion step was then repeated as described above. After the second round of predigestion, 20 mL of HBSS containing 20 mM HEPES (Teknova) was added to the intestinal pieces and incubated for 20 min under continuous rotation followed by vigorous vortexing for 10 s. The pieces were sieved through a 70 μm cell strainer, and the supernatant was washed with a wash solution as described above. All cell fractions contained epithelial cells and were pooled. Intestinal pieces were resuspended in digestion buffer (HBSS with 20 mM HEPES (Teknova), 1.25 mM CaCl₂, 1 mM MgCl₂, 5 mM EDTA (Corning), 5% FCS, Lamina Propria Dissociation Kit enzymes D, R, and A (Miltenyi Biotec), were added according to the manufacturer's protocol) and incubated at 37 °C for 30 min with constant rotation. The intestinal tissue was transferred into a gentleMACS C Tube and processed with a gentleMACS using the manufacturer's program. Perfusion solution was added and the pieces were strained through a 70 μm cell strainer. The supernatant was processed and washed as described above. This second fraction contained the cells of the lamina propria. Intraepithelial and lamina propria fractions of each bat were counted using trypan blue and cells were mixed in a 1:1 ratio for each bat. To prepare a single-cell suspension of the mesentery, the mesentery was tamped through a cell strainer in a Petri dish containing 5 mL of perfusion solution. The number of intestinal and mesenteric cells was set at 1000 cells μL⁻¹ in 1×PBS, and the viability of the samples was determined. A total of 12,000 cells per sample were processed with a Chromium Next GEM Single Cell 5′ Kit v2 (with Dual Index Kit TT Set A and Chromium Next GEM Chip K Single Cell Kit, 10x Genomics) to achieve a recovery of at least 8000 cells. cDNA amplification and libraries were generated according to the manufacturer's instructions and sequenced as multiplexed pools on an Illumina NovaSeq 6000 in paired-end mode (150 cycles, 2 × 150).

## Single-cell downstream analysis

A custom reference genome, appending the A/flat-faced bat/Peru/033/2010 H18N11 sequences (NCBI nucleotide database accession numbers CY125942, CY125943, CY125944, CY125945, CY125946, CY125947, CY125948, and CY125949) to the Jamaican fruit bat genome[80] (Cold Spring Harbor Laboratories genome, NCBI RefSeq assembly GCF_021234435.1), was built with cellranger (v 6.1.2) using mkref function. Then cellranger count function was used to process the fastq files using default parameter settings. Downstream analysis was performed using the Seurat R package (v 4.2.0) running on R 4.2.1[95]. Seurat objects were created from the binary cellranger output (.h5). Samples from the same tissue were merged into a single Seurat object. Both tissues were processed independently but using the same thresholds. Cells meeting the following criteria were selected: number of features (nFeature) between 200 and 3000, maximum read count (nCount) below 20,000, and mitochondrial content below 10%. Genes

quantified in fewer than three cells were excluded. Counts were normalized to the library size with a 10,000-scale factor. Principal component analysis was performed on the top 5000 variable features. Based on the elbow plot, the top 20 principal components were used for nearest neighbor graph construction and Louvain clustering analysis with a resolution of 0.5. Finally, cell and cluster annotations were visualized using the Uniform Manifold Approximation and Projection (UMAP) technique. Enrichment of a particular condition (i.e., uninfected, WT 3 dpi, WT 9 dpi, and ΔNS1 3 dpi, and ΔNS1 9 dpi) within each cluster was quantified using Fisher's exact test.

Cluster annotation was based on differentially expressed human orthologous genes to facilitate the comprehension and cell type annotation as described before[31,32]. Cells were considered infected if at least one of these viral genes was quantified (i.e., read count ≥1): PB1, NP, NA, M, NS1. Cell-type annotation was initially defined based on sctype prediction[96], and cell type marker analysis was performed using MAST[97] with min.pct and logfc.threshold parameters set to 0.25 and 0.1, respectively. Subsequent fine-tuning of cell-type annotation was based on manual marker gene analysis and prior knowledge (Supplementary Data 1).

Epithelial cell types were enterocytes characterized by the expression of *Epcam*, *Creb3l3*, *Fabp1*, *Fabp6*, *Mep1a*, *Apoa4*, and *Tmigd1*[98,99], crypt cells that had enriched levels of *Axin2*, *Olfm4*, *Rgmb*, *Slc12a2*, *Smoc2*, and *Sox9*[98,99] or goblet cells that exhibited characteristic expression of *Agr2*, *Atoh1*, *Clca1*, *Muc2*, *Spdef*, *Tff3*, and *Zg16*[98,99].

Endothelial cells were either vascular endothelial cells (VEC) with a specific expression of *Aplnr*, *Madcam1*, *Pecam1*, *Plvap*, *Tek*, and *Vwf*[99–101] or lymphatic endothelial cells (LEC) characterized by *Flt4*, *Gja4*, *Lyve1*, *Reln*, *Sbspon*, and *Stab2*[99,102].

Undifferentiated hematopoietic stem cells and multipotent progenitors (HSC/MPP) were enriched for *Cd34* and *Id1*[103–106], and progenitor cells (PC) exhibited *Cdk1*, *Mad2l1*, *Tk1*, *Ube2c*, and *Uhrf1*, which are required for cell cycle control and progression.

Enteric glia cell clusters were identified based on previously described marker genes such as *Clu*, *Cryab*, *Igfbp7*, *Mest*, *Nid1*, *Olfml3*, *Postn*, and *Prnp*[99,107].

Myeloid cells clustered into neutrophils (*Cd177*, *Csf3r*, and *Cxcr2*), monocytes (*Ccr2*, *Cd33*, *Cd52*, *Cd300a*, *Lilra5*, and *Sell*), macrophages (*C1qa*, *C1qb*, *Cd68*, *Cd163*, *Il1b*, *Il1rn*, *Il1r2*, and *Lyz*) and *Cd14*⁺ macrophages (*Cd14* and *Slc11a1*), cDC1 (*Batf3*, *Cadm1*, *Clec9a*, *Irf8*, and *Xcr1*), mature lymph node IDC (*Ccr7*, *Fscn1*, *Il4i1*, and *Lamp3*) and mast cells (*Cpa3*, *Fcer1a*, *Gata2*, and *Kit*).

Innate lymphoid cells were present either as type 3 innate lymphoid cells (*Il17a*, *Il17f*, *Il22*, and *Rorc*) or within the mixed NK/T cell cluster that could not be further resolved. NK cells were defined based on the co-expression of the transcription factor *Eomes*, and several killer-cell immunoglobulin-like receptors such as *Klrd1*, *Klrg1*, and *Klrk1*, as well as natural cytotoxicity triggering receptors (*Ncr1* and *Ncr3*). T lymphocytes, including intraepithelial lymphocytes, expressed the T cell receptor (TCR) components *Cd3d* and *Cd3e* and the TCR alpha (*Trav4* and *Trav18*), beta (*Trbv12-4* and *Trbv27*) and gamma chain genes (*Trgv9*), the TCR co-receptor genes *Cd4* and *Cd8a*, and the TCR signaling molecules *Lck* and *Zap70*. Naïve T cells showed additional co-expression of *Ccr7* and *Il7r*.

B lymphocytes were characterized by gene expression of *Cd19*, *Cd79b*, *Cd40*, *Cxcr4*, *Klhl6*, *Pax5*, *Ms4a1*, B cells undergoing affinity maturation and/or class switching expressed high levels of *Aicda*, and plasma cells were enriched for *Jchain*, *Irf4*, and *Tnfrsf17*.

Hallmark gene-set enrichment was performed using Fisher's exact test. Marker genes were selected based on their adjusted $p$ value (<0.05). To increase specificity, we restricted each marker gene list to the top 500 genes based on log₂-fold-change ranking. Gene sets from MSigDB[108] were used as reference. Terms with an adjusted $p$ value below 0.05 were considered significantly enriched.

## Serology

H18N11-specific antibodies were measured by ELISA in bat serum samples collected after euthanasia at 3 and 9 dpi. Briefly, subconfluent MDCK II-MHC-II cells seeded in 96-well plates were treated with 1.5 mg mL$^{-1}$ DEAE-dextran diluted in PBS containing 0.2% BSA (Sigma-Aldrich) and were infected with H18N11 in infection medium or left uninfected. Cells were fixed with 4% PFA (EMD). Starting from a 1:25 dilution in PBS containing 5% NGS (Vector Laboratories, Inc.), serum samples were serially diluted twofold to a final dilution of 1:800 and then added to virus-infected cells. Uninfected cells were used as controls. H18N11-specific antibodies were detected with protein-A/G conjugated to horseradish peroxidase (HRP, Invitrogen) by incubation for 1 h at room temperature. After three thorough washes with PBS containing 0.1% TWEEN 20, followed by three washing steps with PBS, KPL ABTS peroxidase substrate (1-component) (SeraCare Life Science, Inc.) was added, and the absorbance was measured at 405 nm after 5 min. Positive samples were defined as an optical density of 2 standard deviations or greater above the mean of the uninfected cells.

## Histology and immunohistochemistry of small intestines

Formaldehyde-fixed and paraffin-embedded (FFPE) tissue blocks ($n = 5$) contained intestinal specimens from H18N11-infected bats[25] were used to characterize specific immune cell types in the small intestinal mucosa. The sample size comprised three H18N11-infected ($n = 2$ at 3 dpi and $n = 1$ at 6 dpi) and two mock-infected bats. Routine hematoxylin and eosin staining was performed to assess the small intestinal microscopic architecture. Chromogenic immunohistochemistry of FFPE slides was done using an avidin–biotin–peroxidase complex method (Vector Laboratories). The slides were dewaxed for 15 min at 65 °C and then rehydrated using xylenes followed by a standard ethanol gradient. Endogenous peroxidase was blocked by applying 0.5% hydrogen peroxide (Sigma-Aldrich) dissolved in methanol for 30 min at room temperature, followed by washing in TBS buffer. For antigen retrieval, slides were treated in a water bath at 96 °C for 25 min either in citrate buffer at pH 6.0 for detection of wide-spectrum cytokeratin, or TRIS/EDTA buffer at pH 9 for detection of CD20 or Iba1. Blocking of unspecific epitopes was done by using SuperBlock (Thermo Fisher Scientific) for 20 min at room temperature and then incubated overnight at 4 °C in a humid chamber with polyclonal rabbit anti-wide-spectrum cytokeratin (Abcam, 1:2.5), polyclonal rabbit anti-CD20 (Thermo Fisher Scientific, 1:2.5) or polyclonal rabbit anti-Iba1 (Novus Biologicals, 1:18.5). The slides were washed two times with TBS buffer, and were subsequently incubated with biotinylated goat anti-rabbit IgG (H + L) antibody (Vector Laboratories) for 30 min at room temperature, followed by the avidin–biotin–peroxidase complex method (Vector Laboratories) using 3-amino-9-ethylcarbazol (AEC, medac diagnostika) as a chromogen and hematoxylin as counterstain. The stained slides were mounted with Aquatex® (VWR).

To obtain chromogenic brightfield images an Olympus BX 53 (Olympus Life Sciences) equipped with 4x/0.13, 10x/0.30, 20x/0.50, 40x/0.75, and 100x/1.30 oil-immersion objectives, a 5-megapixel digital camera (DP26, Olympus Life Sciences) and cellSens (v.1.18, Olympus kit solution) was used. Photomicrographs were normalized in white balance and detail sharpening was applied with the unmasking filter in Photoshop 2023 (Adobe).

## In situ hybridization and immunofluorescence

Dual staining of H18-vRNA and Iba1$^+$ macrophages was done by sequential in situ hybridization (ISH) using RNAscope® 2.5 HD Detection Kit-RED (Advanced Cell Diagnostics) and immune-fluorescence on FFPE slides according to the manufacturer's instructions (MK 51-149/ Rev B/Date 10/05/2020) with minor modifications. FFPE slides of the small intestine were deparaffinized as described before, endogenous peroxidase was blocked with ready-to-use H$_2$O$_2$ (RNAscope® 2.5

Pretreat Reagents, Advanced Cell Diagnostics) and slides were washed in distilled H$_2$O. Target retrieval was done by incubating slides with RNAscope co-detection target retrieval solution (Advanced Cell Diagnostics) at 98–102 °C for 15 min in a steaming oven, followed by rinsing in distilled H$_2$O. Next, we performed antigen retrieval by incubating the slides in a water bath at 96 °C for 25 min at pH 6.0 in citrate buffer followed by tissue permeabilization with 0.1% Triton X-100 (Sigma-Aldrich) dissolved in PBS for further 10 min. Tissues on slides were outlined with a hydrophobic barrier pen, and unspecific epitopes were handled by using a co-detection blocker (Advanced Cell Diagnostics) for 1 h. After rinsing the slides with PBS-T, the polyclonal anti-Iba1 antibody (Novus Biologicals, 1:500) to detect macrophages and a polyclonal antibody targeted against an irrelevant epitope (*Helicobacter pylori*, Dako, 1:500) diluted in co-detection antibody diluent (Advanced Cell Diagnostics) were incubated at 4 °C overnight. On the next day, slides were first washed in PBS-T, then fixed in 3.7% neutral buffered formaldehyde to shield the antibody-bound epitopes, followed by a second washing step in PBS-T before digestion of tissues with kit-provided proteinase (Advanced Cell Diagnostics) for 30 min at 40 °C. In situ hybridization was performed to detect H18-vRNA using the RNAscope 2.5 HD Reagent Kit-RED (Advanced Cell Diagnostics) together with the HybEZ II Hybridization System (Advanced Cell Diagnostics) with a 20ZZ probe against the HA segment of A/flat-faced bat/Peru033/2010(H18N11) (Advanced Cell Diagnostics, base pairs 26–1132, GenBank accession number: CY125945.1)[25]. Negative (RNA-Scope® Negative Control Probe - DapB – *Bacillus subtilis*, Advanced Cell Diagnostics) and positive controls (RNAScope® Positive Control Probe – peptidylprolyl isomerase B (PPIB) of Jamaican fruit bat (predicted), Advanced Cell Diagnostics) were included in each run. As controls, the following combinations were used in a dual labeling approach: (i) an irrelevant probe directed against dapB of *Bacillus subtilis* strain SMY and a primary antibody directed against *Helicobacter pylori*, respectively, (ii) a probe directed against PPIB of Jamaican fruit bat in combination with Iba1 to detect macrophages. Tissue slides were exposed to ISH target probe pairs and incubated at 40 °C in a hybridization oven for 2 h. After rinsing in RNAscope® Wash Buffer (Advanced Cell Diagnostics), ISH signals were amplified using a kit-provided pre-amplifier and amplifier (RNAscope® 2.5 HD Detection Kit, Advanced Cell Diagnostics) conjugated to alkaline phosphatase and incubated with a Fast Red substrate solution for 10 min at room temperature. Tissue slides were washed three times in PBS-T before applying the secondary Alexa Fluor® 488 donkey anti-rabbit antibody (Dianova; 1:500) for 30 min at room temperature. Slides were again washed three times in PBS-T before adding VECTASHIELD Vibrance® Antifade Mounting Medium (Vector Laboratories) and coverslipped with high precision glass slides No. 1.5 H, 170 ± 5 µm (Marienfeld).

## Confocal laser scanning microscopy, deconvolution, 3D-image analysis

Small intestinal tissue slides ($n = 2$) with Iba1$^+$ macrophages and H18-vRNA were imaged with the confocal laser scanning microscope (CLSM) Leica TCS SP8 and the Leica Application Suite X (LAS-X 3.5.7) software applying a bidirectional scan with a scan speed of 600 Hz, pinhole 1 airy unit, and 4x line averaging. Each of the channels was separately scanned. Alexa Fluor® 488 (channel 1) was excited at 496 nm (PMT detection range 502–536 nm) and FastRed (channel 2) at 561 nm (HyD detection range 567–635 nm). The transmission of each laser line was set so that the signal (8-bit encoding) was just below saturation. Using an APO 63x/1.20 water objective and the Leica stage navigator software, an overview image (tile scan) was created to identify appropriate locations for subsequent scanning of image stacks. Applying Huygens criteria for excitation at 496 nm (voxel size: 49 × 49 × 163 nm), image stacks with up to 56 layers were recorded at selected locations.

Deconvolution of CLSM images, using Huygens Professional (v 22.10, SVI), was applied to reduce optical aberrations. Microscopic

parameters were set according to the image metadata. The refractive indices (IR) of the embedding media was 1.45 and of the immersion media (water) was 1.33. A theoretical point spread function was used, and the following deconvolution parameters were set for processing with the Classic Maximum Likelihood Estimation (CMLE) algorithm: signal-to-noise ratio = 24, absolute background for both channels = 5, maximum iterations = 200 acuity for channel 1 (Alexa Fluor® 488) = auto, for channel 2 (Fast Red) = 100, quality threshold = 0.001.

Imaris (v.9.9.1, Oxford Instruments) was used for 3D imaging of deconvolved CLSM data. Surface objects (surfaces) representing Alexa Fluor® 488 labeled macrophages were created by voxel segmentation using the following creation parameter: surface grain size = 0.1 μm, absolute intensity = 16.3, number of voxels above 10. Fast Red-labeled H18-vRNA was imaged by segmentation of voxels of a defined intensity as classified spherical objects (spots) using the following creation parameter: estimated XY diameter = 0.16 μm, background subtraction, quality threshold above 24.3, classification criteria: shortest distance to surface (Iba1$^+$cell membranes) = 0.054 μm.

## Reporting summary

Further information on research design is available in the Nature Portfolio Reporting Summary linked to this article.

## Data availability

The scRNA-seq data generated in this study has been deposited in NCBI's Gene Expression Omnibus and accessible through GEO Series accession number GSE243982. This paper does not report original code. Source data are provided with this paper.

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

## Acknowledgements

We thank P. Aichele, B. Bengsch, M. Hofmann, and Y. Tanriver for their excellent support and immunological expertise, E. Quente, H. Gräfe for their technical support and advice, R. Albrecht and R. Cadagan for logistical help and shipment, and G. Chase for assistance with manuscript preparation. The work was funded by grants from the European Research Council (ERC) to M.S. (NUMBER 882631—Bat Flu), the National Institute of Allergy and Infectious Diseases (NIAID) to T.S. (R01 AI140442), and to W.M. and T.S. (R01 AI134768), and by the Medical Faculty, University of Freiburg through the "Hans A. Krebs Medical Scientist Program" to K.C. It was also supported by the following grants: Deutsche Forschungsgemeinschaft (DFG) within the CRC1160 (project ID 256073931-Z02), CRC/TRR167 (project ID 259373024-Z01), CRC1453 (project ID 431984000-S1), and CRC1479 (project ID 441891347-S1) to M.Bo., the German Federal Ministry of Education and Research (BMBF) within the MIRACUM Consortia FKZ 01ZZ1801B to M.Bo., PM4Onco–FKZ 01ZZ2322A to M.Bo., and EkoEstMed–FKZ 01ZZ2015 to G.A.

## Author contributions

Conceptualization: M.S., T.S., and K.C.; bat experiments: B.B., S.Z., C.R., and D.T.F.; scRNA-seq experiments: T.S.D.; bioinformatical analysis: G.A. and M.Bo.; histopathology: J.S., J.K., and R.U.; human immune cell infections: S.K., L.H., J.St., J.F.V.O., T.C., and M.R.; reagents and expertise: M.K.O., M.H.T., W.M., and M.Be.; writing – original draft: K.C.; writing—review and editing: S.K., M.S., P.R., and T.S.; supervision and project administration: T.S. and K.C.; funding acquisition: G.A., M.S., T.S., and K.C.

## Funding

## Competing interests

The authors declare no competing interests.

## Additional information

[1]Institute of Virology, Medical Center University of Freiburg, Freiburg, Germany. [2]Faculty of Medicine, University of Freiburg, Freiburg, Germany. [3]Center for Vector-borne Infectious Diseases, Department of Microbiology, Immunology and Pathology, College of Veterinary Medicine and Biomedical Sciences, Colorado State University, Fort Collins, CO, USA. [4]Institute of Medical Bioinformatics and Systems Medicine, Medical Center - University of Freiburg, Freiburg, Germany. [5]Institute of Veterinary Pathology, Faculty of Veterinary Medicine, Leipzig University, Leipzig, Germany. [6]BioImaging Core Facility, Faculty of Veterinary Medicine, University of Leipzig, Leipzig, Germany. [7]Department of Microbiology, Immunology and Pathology, Colorado State University, Fort Collins, CO, USA. [8]Spemann Graduate School of Biology and Medicine, University of Freiburg, Freiburg, Germany. [9]Faculty of Biology, University of Freiburg, Freiburg, Germany. [10]Department of Rheumatology and Clinical Immunology, Medical Center - University of Freiburg, Faculty of Medicine, University of Freiburg, Freiburg, Germany. [11]Center for Chronic Immunodeficiency (CCI), Medical Center - University of Freiburg, Freiburg, Germany. [12]Institute for Transfusion Medicine and Gene Therapy, Medical Center-University of Freiburg, Freiburg, Germany. [13]Department of Veterinary Pathobiology and Department of Molecular Microbiology and Immunology, University of Missouri, Columbia, MO, USA. [14]German Cancer Consortium (DKTK), Partner site Freiburg, a partnership between DKFZ and Medical Center - University of Freiburg, Freiburg, Germany. [15]CIBSS – Centre for Integrative Biological Signalling Studies, University of Freiburg, Freiburg, Germany. [16]Division of Clinical and Experimental Immunology, Institute of Immunology, Center for Pathophysiology, Infectiology and Immunology, Medical University of Vienna, Vienna, Austria. [17]Institute of Diagnostic Virology, Friedrich-Loeffler-Institut, Greifswald, Insel Riems, Germany. [18]These authors contributed equally: Susanne Kessler, Bradly Burke. ✉e-mail: Tony.Schountz@colostate.edu; kevin.ciminski@uniklinik-freiburg.de

