## [Peer Review File · Nature Communications]

Deciphering bat influenza H18N11 infection dynamics in male Jamaican fruit bats on a single-cell levelEditorial Note: This manuscript has been previously reviewed at another journal. This document only contains reviewer comments and rebuttal letters for versions considered at *Nature Communications*.

REVIEWER COMMENTS

Reviewer #1 (Remarks to the Author):

The authors have addressed all the concerns. This study will add useful information towards understanding how reservoir bat species tolerate virus infections.

Reviewer #2 (Remarks to the Author):

In this resubmission, Kessler et al. address several issues highlighted in the previous review cycle. The authors present a study that investigates an interesting model of zoonotic flu infection. While currently of specific interest to researchers working with the Jamaican Fruit Bat, with further development, this work has the potential to impact a broader field. The single-cell map of the bat's mesentery and intestine promises to be a valuable resource for this niche community.

The paper focuses on the ability of the H18N11 virus to productively infect immune cells, particularly macrophages. Previous extensive research, as detailed in [e.g. <https://www.ncbi.nlm.nih.gov/pmc/articles/PMC5725990/table/T1/?report=objectonly>], has established that some flu strains can infect sometimes productively antigen-presenting cells and macrophages. The study's observation that H18N11 can productively infect bat and human macrophages, among other immune cells, is intriguing. However, the broader implications of this finding should be clarified for a general audience.

Regarding the methodology, my initial interpretation of "pooled scRNA-seq data" led to confusion, as it seemed to imply that all cells were processed together in the 10x GEM. Upon reevaluation of the methods section, I understand each sample was processed individually. I recommend revising the text for clarity to avoid such misunderstandings. Additionally, the manuscript should correct the statement that "UMAP revealed 24 distinct cell clusters" to accurately reflect that clustering was performed with Louvain clustering on the 20 principal components, not on the UMAP itself.

Major Points:

The absence of a table listing per-sample cell numbers and cell type frequencies limits the interpretability of the analyses presented.

Figure 1c/d could potentially be misleading. The apparent tenfold increase in B cells in WT infections versus NS1 KO infections is not as evident in the mesentery (Figure 1d), possibly influenced by the difference in the total cell counts available per sample. Supplementary Figure 5 (incorrectly cited as Figure 3 in the text) is challenging to decipher, and the exact nature of the statistical test conducted is unclear—was it a Fisher exact test comparing each set of replicates by condition and uninfected status?

Statements like "the number of cDC1s and both macrophage subsets were enriched in Δ NS1-infected bats at 3 dpi compared to uninfected controls" could be more helpful. The trend here is ambiguous, and attributing significance to a single timepoint where the data appears noteworthy may be misleading. For instance, the fluctuating proportions of cDC1 cells described in Figure 1c may be nothing more than natural variability, not a meaningful difference.

Concerning the differential expression analysis, the manuscript states, "we performed hallmark gene set analysis on intestinal and mesenteric cells from infected bats compared to uninfected control animals." However, the criteria used to rank genes in the GSEA are not specified, nor is there a reference explaining what constitutes hallmark gene set enrichment analysis, apart from its application to predefined sets in PMCID: PMC4707969. Questions arise when gene sets like Oxidative Phosphorylation and Adipogenesis show upregulation in the mesentery but

downregulation in the intestine when comparing 9dpi to 3dpi for both WT and Δ NS1 infections. How can these gene sets exhibit significant, yet opposite, changes in mesentery versus intestine?

It is unclear why gene set enrichment analysis was performed at the bulk level rather than at the cell type or meta-cell type level, such as lymphocytes or immune cells.

Minor Points:

Figure 4 may be potentially misleading. In the intestine, while there is a slight enrichment of infected immune cells, the actual number of infected cells compared to the large total count results in a significant yet modest p-value, indicating an overall low infection rate. In the mesentery, the predominance of immune cells profiled may explain why all 26 infected cells are immune cells, which could suggest a low general infection rate for H18N11, potentially with a marginal preference for immune cells in the intestine.

In Figure 1, the 'unknown' cluster in the mesentery potentially connects many other clusters, which is often indicative of artifacts like doublets or empty droplets. It would be beneficial if the authors addressed whether this unknown cluster has been checked for such anomalies.

Reviewer #4 (Remarks to the Author):

The authors have addressed previous comments:

- MHC-II expression of MDCK II cells used for virus growth and titration.
- Uninfected control bat sera in ELISA serology
- Fecal shedding of infectious virus.

Overall, this manuscript reports pioneering research approaches and findings that add substantial new knowledge to our current understanding of the natural history of H18N11 in bats.

Reviewer #2 (Remarks to the Author):

In this resubmission, Kessler et al. address several issues highlighted in the previous review cycle. The authors present a study that investigates an interesting model of zoonotic flu infection. While currently of specific interest to researchers working with the Jamaican Fruit Bat, with further development, this work has the potential to impact a broader field. The single-cell map of the bat's mesentery and intestine promises to be a valuable resource for this niche community.

The paper focuses on the ability of the H18N11 virus to productively infect immune cells, particularly macrophages. Previous extensive research, as detailed in [e.g. <https://www.ncbi.nlm.nih.gov/pmc/articles/PMC5725990/table/T1/?report=objectonly>], has established that some flu strains can infect sometimes productively antigen-presenting cells and macrophages. The study's observation that H18N11 can productively infect bat and human macrophages, among other immune cells, is intriguing. However, the broader implications of this finding should be clarified for a general audience.

We show that H18N11 can infect not only bat immune cells but also human immune cells, especially macrophages. The broader implications of these results, such as the zoonotic potential or the influence on further surveillance studies, are already discussed at some length in the discussion section of the present manuscript (lines 420-428). Further discussion or interpretation would be excessively speculative based on our available data.

Regarding the methodology, my initial interpretation of "pooled scRNA-seq data" led to confusion, as it seemed to imply that all cells were processed together in the 10x GEM. Upon reevaluation of the methods section, I understand each sample was processed individually. I recommend revising the text for clarity to avoid such misunderstandings.

To improve the clarity of our text and to avoid confusion about our working procedure, we have removed the word "pooled". We also now emphasize that the intestine and mesentery of each bat were sequenced and analyzed individually.

*Line 118-120: "At 3 and 9 dpi, representing an early and a late time point of the acute infection²⁵, we euthanized three animals from each infection group and generated single-cell transcriptomes **for each** bat intestine and mesentery (Fig. 1a)."*

Line 122-125: "To maximize the identification of rare cell populations and to generate a complete immunological landscape of the intestine and mesentery, we analyzed scRNA-seq data for these organs from both infected and uninfected bats."

Additionally, the manuscript should correct the statement that "UMAP revealed 24 distinct cell clusters" to accurately reflect that clustering was performed with Louvain clustering on the 20 principal components, not on the UMAP itself.

We thank the reviewer for pointing out our imprecise wording. UMAP analysis is not used to reveal cell clusters, but rather is an approach to visualize these cell clusters. As suggested by the reviewer, we have revised the text as follows:

*Line 126-131: "**In total, 24 distinct cell clusters were identified using Louvain clustering on the top 20 principal components and visualized using uniform manifold approximation and projection (UMAP) (Fig. 1b and Supplementary Fig. 3a). These intestinal cell clusters were manually annotated based on the RNA expression profile of highly conserved mammalian marker genes previously reported in the literature for human and mouse (Supplementary Fig. 4 and Supplementary Table 1)**^{36,37}."*

Line 171-174: "In total, 18 cell clusters were identified using Louvain clustering on the top 20 principal components and visualized using UMAP (Fig. 1d and Supplementary Fig. 5a). These mesenteric cell clusters were manually annotated using canonical cell expression markers (Supplementary Fig. 6 and Supplementary Table 1)."

Major Points:

The absence of a table listing per-sample cell numbers and cell type frequencies limits the interpretability of the analyses presented.

We agree with the reviewer's suggestion to show the number of cells and the frequency of cell types in each sample, on which Supplementary Figures 3 and 5 are based. We now show the per-sample cell numbers and cell type frequencies in the updated Supplementary Table 2.

Figure 1c/d could potentially be misleading. The apparent tenfold increase in B cells in WT infections versus NS1 KO infections is not as evident in the mesentery (Figure 1d), possibly influenced by the difference in the total cell counts available per sample. Supplementary Figure 5 (incorrectly cited as Figure 3 in the text) is challenging to decipher, and the exact nature of the statistical test conducted is unclear—was it a Fisher exact test comparing each set of replicates by condition and uninfected status?

The bar graphs in Figure 1c and e show the percentages of the different immune cell types relative to all immune cells in the intestine or the mesentery for each infection group (uninfected, WT, and Δ NS1) at 3 and 9 dpi. To ensure full transparency of the data, as requested by the reviewer in the last round of review, we have already added sample-wise statistics for each cell type in both intestine (Supplementary Figure 3b) and mesentery (Supplementary Figure 5b), and now also show a table listing the per-sample cell numbers (Supplementary Table 2). In addition, to improve clarity of the text, we have i) deleted the phrase "the number of" in lines 161-169 because the data are presented as percentages, and ii) rephrased the Figure legends of Figures 1 c and e.

*Lines 1106-1108: "Bar graph showing the proportion of **the indicated** immune cell types relative to all immune cells in the intestine of uninfected, WT H18N11, or Δ NS1-infected bats at the indicated time points."*

*Lines 1109-1111: "Bar graph showing the proportion of **the indicated** immune cell types relative to all immune cells in the mesentery of each bat group at the indicated time points."*

We performed Fisher's exact test to evaluate whether a cell type was enriched in either the intestine or mesentery of a given bat compared to all other bat samples, which compares each sample to all other samples of an organ (e.g., intestinal cells from all animals in all infection groups). This takes into account the number of cells in each sample as well as the distribution of cells among different cell types. The odds ratio shown in the figure indicates whether a sample has an over- or under-representation of specific cell types compared to all other samples within the same tissue. This statistical analysis enhances the interpretative power of Figures 1c and e.

Statements like "the number of cDC1s and both macrophage subsets were enriched in Δ NS1-infected bats at 3 dpi compared to uninfected controls" could be more helpful. The trend here is ambiguous, and attributing significance to a single timepoint where the data appears noteworthy may be misleading. For instance, the fluctuating proportions of cDC1 cells described in Figure 1c may be nothing more than natural variability, not a meaningful difference.

The reviewer refers to a section of the manuscript where we precisely and carefully describe the significant enrichment of certain cell types in the bat samples after infection (see comment

above regarding the statistics). The cell types cDC1, macrophages, and Cd14⁺ macrophages mentioned by the reviewer are indeed enriched in the intestine of all Δ NS1-infected bats at 3 dpi compared to all other samples. We must therefore disagree with the reviewer's statement, as the enrichment of these cell types represents a consistent and significant trend across three different bat intestinal samples. This finding should not be taken as ambiguous, but rather as compelling evidence of a genuine biological response to viral infection.

Concerning the differential expression analysis, the manuscript states, "we performed hallmark gene set analysis on intestinal and mesenteric cells from infected bats compared to uninfected control animals." However, the criteria used to rank genes in the GSEA are not specified, nor is there a reference explaining what constitutes hallmark gene set enrichment analysis, apart from its application to predefined sets in PMCID: PMC4707969.

As stated in our prior rebuttal, we did not use Gene Set Enrichment Analysis (GSEA) (which is sometimes used as a general term for gene set enrichment) in our study. Instead, we performed gene set enrichment (GSE) using the Fisher test, which does not require a ranked gene list. We provide a list of up- and down-regulated genes as described in the Methods section. After determining adjusted p-values (< 0.05), we selected the top 500 genes based on log₂FC for further analysis.

The hallmark gene set used in our analysis was derived from MSigDB, as described in the Methods section. Recognized as a prominent resource in the scientific community, MSigDB was recently highlighted as one of the "Common Databases" in a Nature Reviews Genetics article by Heumos et al. <https://pubmed.ncbi.nlm.nih.gov/37002403/>. Within MSigDB, the hallmark collection stands out for its refined gene sets, carefully curated to minimize redundancy between terms while encapsulating the most relevant biological information from broader gene sets. By choosing the hallmark gene set, we aimed to ensure clarity and precision in our analysis, avoiding the potential information overload associated with larger and more redundant databases such as Gene Ontology.

We have added a reference in the Methods section that cites the study that established the "hallmark" gene sets.

Line 678-679: "Gene sets from MSigDB¹⁰⁸ were used as reference."

Questions arise when gene sets like Oxidative Phosphorylation and Adipogenesis show upregulation in the mesentery but downregulation in the intestine when comparing 9dpi to 3dpi for both WT and Δ NS1 infections. How can these gene sets exhibit significant, yet opposite, changes in mesentery versus intestine?

The reviewer is correct that certain hallmark gene sets, such as "fatty acid metabolism" or "oxidative phosphorylation", are differentially regulated in the intestine compared to the mesentery as shown in Figure 2 a and b. Because most cells in the intestine are non-immune cells, whereas the majority of cells in the mesentery are immune cells (see Figure b and d), it is possible that the differences mentioned by the reviewer are due to the different response of the cell types to infection with H18N11. Unfortunately, a comprehensive analysis of the individual hallmark gene sets for each infection group at each time point is beyond the scope of this study (see response below). However, we have revised the text to provide a possible explanation for the differential regulation of these hallmark gene sets.

Line 222-225: "**Because most of the cell types in the intestine were epithelial and other non-immune cells, whereas the vast majority of the mesenteric cell types were immune cells (Fig. 1 b,d and Supplementary Table 2), it is possible that these differences in the regulation of the metabolic pathways are due to a distinct transcriptional responses of the cell types to H18N11 infection.**"

It is unclear why gene set enrichment analysis was performed at the bulk level rather than at the cell type or meta-cell type level, such as lymphocytes or immune cells.

We performed a pseudo-bulk analysis to provide a global overview of the regulation of the hallmark gene sets in the intestine and mesentery between the different infection groups. This is actually in line with the previous request by the reviewer to “show what’s globally changed between the two [i.e. WT and Δ NS1 infection groups] in order to conclude on how the host responds to the viral infection.” This hallmark gene set analysis showed that the IFN type I and II response was globally upregulated at 3 days post infection in both the intestine and mesentery. As we considered the regulation of the IFN response to be most relevant for our study, we provide further details on the regulation of individual genes belonging to the IFN pathway in each cell type (Figure 2c, d; Figure 3, Supplementary Figure 7). We have chosen not to include a hallmark gene set analysis for each cell type of each infection condition and time point, as this would overload the manuscript with data and more than 25 additional figures, likely disrupting the reading experience without adding more insight or value to the particular aspects addressed in the manuscript. However, as the data will be publicly available upon publication of the manuscript, readers interested in other aspects of these pathways are welcome to explore this further.

Minor Points:

Figure 4 may be potentially misleading. In the intestine, while there is a slight enrichment of infected immune cells, the actual number of infected cells compared to the large total count results in a significant yet modest p-value, indicating an overall low infection rate. In the mesentery, the predominance of immune cells profiled may explain why all 26 infected cells are immune cells, which could suggest a low general infection rate for H18N11, potentially with a marginal preference for immune cells in the intestine.

No statistical tests were performed to determine the significant infection frequency of specific cell types, contrary to the assertion in this comment. As mentioned in the last revision and clearly shown in Figure 4 of the manuscript, this is a purely qualitative description of the cell types infected by H18N11. Moreover, we clearly state in the manuscript text that H18N11 transcripts were found in intestinal epithelial cells and leukocytes. Therefore, we must disagree with the reviewer's statement that Figure 4 is misleading.

In Figure 1, the 'unknown' cluster in the mesentery potentially connects many other clusters, which is often indicative of artifacts like doublets or empty droplets. It would be beneficial if the authors addressed whether this unknown cluster has been checked for such anomalies.

We appreciate the reviewer's comment regarding the quality of our data set. As part of our rigorous quality control process, we have taken steps to remove cells with excessively high read counts or an unusually high number of features to reduce the risk of potential duplicates. Following the reviewer's suggestion, we examined the distribution of read and feature counts per cluster, which showed a slightly lower count in cluster 2 compared to others. Therefore, we acknowledge the possibility that this particular cluster may be enriched in empty droplets and have revised the text accordingly to account for this scenario. Importantly, this cluster was not included in any further analysis.

*Lines 174-179: “Non-immune cell types in the mesentery included progenitor cells (PCs), enterocytes, and undifferentiated hematopoietic stem cells and multipotent progenitors (HSC/MPPs), as well as a population of cells (**cluster 2**) that exhibited an expression profile that could not be attributed to a specific cell type. **Because this unknown cell cluster also had relatively low read and feature counts, suggesting a potential overpopulation of empty droplets, we excluded these cells from further analysis.**”*

REVIEWERS' COMMENTS

Reviewer #2 (Remarks to the Author):

The reviewers addressed the points we made satisfactorily. I do not have further comments.